# JET EXPANSIONS OF RESIDUAL COMPUTATION

## ABSTRACT

We introduce a framework for expanding residual networks using *jets*, operators that generalize truncated Taylor series. Our method provides a systematic approach to disentangle contributions of different computational paths to model predictions. In contrast to existing techniques such as distillation, probing, or early decoding, our expansions rely solely on the model itself and requires no data, training, or sampling from the model. We demonstrate how our framework grounds and subsumes the logit lens, reveals a (super-)exponential path structure in the network depth and opens up several applications. These include the extraction of $n$-gram statistics from a transformer large language model, and the definition of data-free toxicity scores. Our approach enables data-free analysis of residual networks for model interpretation, development, and evaluation.

## 1 INTRODUCTION

Machine learning models, particularly large-scale foundation models, have become increasingly prevalent and impactful across a wide range of domains (Wei et al., 2021; Bommasani et al., 2023; Touvron et al., 2023b). While delivering strong results, their black-box nature has led to the development of techniques to assess their behavior and gain insights into their internal mechanisms. In this space, mechanistic interpretability (MI) (see e.g. Bereska & Gavves, 2024; Ferrando et al., 2024, for recent surverys) has emerged as an alternative to more classic local attribution methods such as SHAP (Lundberg, 2017) or integrated gradient (Sundararajan et al., 2017). Contrary to these methods, which seeks to trace output behavior back to the network input, MI focuses on tracing behavior back to the model itself. It seeks to uncover learned "algorithms" that are embedded in the model weights and computational structure, with the aim of developing a global understanding of – and, ultimately, to reverse engineer – neural computation.

The great majority of MI work uses a hypothesis-and-dataset-driven approach (see for example Goldowsky-Dill et al. (2023)), in that it first formalizes a hypothesis, then chooses or curates a dataset to probe the model, it applies techniques such as path patching (Wang et al., 2022) or causal tracing (Meng et al., 2022), and then possibly refines the initial hypothesis. While this approach to MI is valuable, it can limit the ability to perform open-ended exploration-driven studies aimed at uncovering global behavior and charting "maps" that connect computation to behavior. In this regard, studies such as Veit et al. (2016) or Elhage et al. (2021) focus on the intrinsic computation that is carried out by a model, offering complementary views to the hypothesis-and-dataset-driven approach. Yet, these studies often make unrealistic assumptions of the model, making it unclear how much of the derived understanding can be transferred to real-world models and applications.

This paper contributes to this latter direction, presenting a general-purpose framework to manipulate a residual computational graph with the aim of decomposing it into individual input-to-output computational paths, which we can then further analyze to extract behaviors. Our method is based on the simple observation that we can recursively expand a residual computational graph by selectively applying *jet operators* (Ehresmann, 1951), which one can think of as the functional counterpart of truncated Taylor series. This process, which we call the *jet expansion* of a model, gives rise to a class of equivalent functional rewritings of the original network into the sum of polynomial terms (that we see as input-to-output functions and dub *jet paths*) and non-linear remainders. The framework does not make particular assumptions on the input model and, as it operates in the space of functions (rather than function evaluations), it requires no input data. For transformer language models, we show how specific instantiations linked to $n$-gram models make it feasible to exhaustively evaluate the jet paths over the entire input space, enabling end-to-end data-free global interpretability.

We focus on residual networks (He et al., 2016), particularly transformers (Vaswani et al., 2017), operating at the granularity of residual blocks (e.g., self-attention or MLP blocks). This approach simplifies our presentation, aligns with (Veit et al., 2016), and remains relevant given the ubiquity of residual computation in practice. In section 4, we describe several instantiations of our framework, some encompassing previously proposed interpretability tools like LogitLens (nostalgebraist, 2021b). Based on these instantiations, we present extensive case studies on auto-regressive large language models (LLMs) from varying families and sizes, including *GPT*, *Llama* and *OLMo*. Our findings demonstrate that jet expansion offers a versatile toolkit – jet lens, jet paths and jet $n$-grams – for interpreting LLMs: i) analyzing their inner working (section 5.2); ii) debugging pretraining dynamic (section 5.3); and iii) examining fine-tuning effects (section 5.4), contributing to more transparent and responsible LLM usage. We conclude with a discussion about potential future research directions that this work opens, alongside its current limitations.

## 2 RESIDUAL NETWORKS AND THEIR REWRITINGS

We start by reviewing the archetypal computational structure of residual networks and discuss the case of linear residual networks as a canonical example of functions that are intrinsically expanded.

**Residual networks.** We focus on network architectures whose main body consists of multiple recursive residual blocks, while the input and output are managed respectively by an encoding and a decoding module. Let $\mathcal{Z}$ be an input space (e.g., sequences of tokens), $c \in \mathbb{N}^+$ be the number of classes (e.g., a vocabulary size), $\mathcal{Y} = \mathbb{R}^c$ be a space of output logits and $d \in \mathbb{N}^+$ be a hidden dimension. Formally, we are concerned with functions $q : \mathcal{Z} \to \mathcal{Y}$ described as follows:

$$q = \upsilon \circ h_L, \quad \text{where } h_L : \mathcal{Z} \to \mathbb{R}^d, \quad h_L = \bigcirc_{l=1}^{L} \beta_l \circ \eta, \tag{1}$$

where $L \in \mathbb{N}^+$ is the number of residual blocks (e.g. recursive depth), $\eta : \mathcal{Z} \to \mathbb{R}^d$ is an input encoding module (e.g. token embedding layer), $\bigcirc$ denotes repeated functional composition, and

$$\beta_l : \mathbb{R}^d \to \mathbb{R}^d \quad \text{for } l \in [L] \qquad \beta_l = \text{id} + \gamma_l, \qquad \gamma_l : \mathbb{R}^d \to \mathbb{R}^d, \tag{2}$$

$$\upsilon : \mathbb{R}^d \to \mathcal{Y} \qquad \upsilon(x) = U\,\gamma_{L+1}(x) \qquad U \in \mathbb{R}^{c \times d}, \gamma : \mathbb{R}^d \to \mathbb{R}^d, \tag{3}$$

are respectively residual blocks with nonlinearities $\gamma_l$'s (e.g., input-normalized causal self-attentions or MLPs), and the output decoding module (e.g., an unembedding projection $U$ after a layer normalization $\gamma_{L+1}$); id is the identity map. We leave all parameters *implicit* and assume all functions are $C^\infty$. Optimized for classification (e.g., next token prediction for autoregressive language models), the function $q$ outputs unnormalized conditional probabilities (or logits) in that $\mathbb{P}_q(\text{"}z \text{ belongs to class } i\text{"}|z) = \text{Softmax}[q(z)]_i$, for $z \in \mathcal{Z}$. In residual networks, the recursive links allow the "storage" of computation from all previous layers and the embedded input, leading to an accumulation of information across depths. This is highlighted by unrolling the computation of eq. (1) up to a block $l \in [L]$, setting $h_0 = \eta$:

$$h_l = \bigcirc_{j=1}^{l} \beta_j \circ \eta = \eta + \sum_{j=1}^{l} \gamma_j \circ h_{j-1}; \quad q = \upsilon \circ \eta + \sum_{l=1}^{L} \upsilon \circ \gamma_l \circ h_{l-1} \tag{4}$$

Elhage et al. (2021) introduces the term *residual stream* to describe $h_l$, a concept that can be traced back to Hochreiter & Schmidhuber (1997) and Srivastava et al. (2015). Veit et al. (2016) describe and study the unrolled structure of the final residual stream $h_L$, which reveals a number of paths from the input to the decoder that grows *linearly* with the network depth.

**Linear residual networks.** The presence of non-linearities at each block (and at the decoding module) prevents us from *directly* expanding the input-to-output computation further.[1] Linear residual networks do not have this impediment. Indeed, if $\gamma_i(x) = A_i x$ for some $A_i \in \mathbb{R}^{d \times d}$, $\eta = E$ and $\gamma = \text{id}$, we have that

$$q = U\left(\sum_{S \in 2^{[L]}} \prod_{j \in S} A_j\right) E = \sum_{S \in 2^{[L]}} q_S \tag{5}$$

where $2^{[L]}$ is the power set of $[L] = \{1, \ldots, L\}$ and the $q_S = U(\prod_{j \in S} A_j)E = UW_S E$, with $W_\emptyset = I$. Equation (5) writes ("expands") the linear network into a combination of $2^L$ input-to-output paths $q_S : \mathcal{Z} \to \mathcal{Y}$, themselves linear functions. This enables a detailed analysis of each path's contributions (e.g. one may look at the norm of each $W_S$ as a measure of global path importance), roles, and interactions, as well as understanding global input-output relationships.

---

[1] One can still recover an exponential expansion of gradient paths when considering $\nabla q$, e.g. to analyze behavior during training, as Veit et al. (2016) do. In this work, however, we solely focus on the forward dynamic of the network.

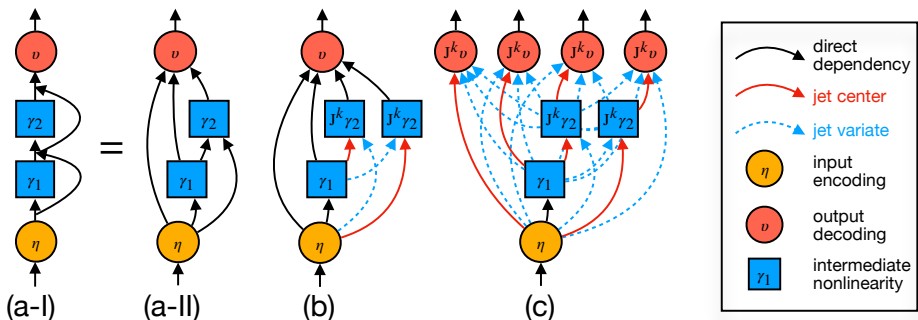

Figure 1: Representation of a two-blocks residual net (a, a-bis) and its exponential expansion steps (b, c).

# 3 RECURSIVE EXPANSION OF RESIDUAL NETWORKS WITH JETS

To tackle non-linearities and enable expansions in general residual networks similar to that of eq. (5), we turn to jets (Ehresmann, 1951)In this section, we first introduce key concepts pertaining jets and then move to describe the general algorithm for expanding residual computation.

**Jet operators and their convex combinations** We recall that, for a function $f \in C^{k+1}(\mathbb{R}^d, \mathbb{R}^d)$ and $x, y \in \mathbb{R}^d$, Taylor's theorem asserts that

$$f(y) = f(x) + \sum_{j=1}^{k} (j!)^{-1} D^j f(x)(y-x)^{\otimes j} + O(\|y-x\|^{k+1}) \tag{6}$$

where $x, y$ are respectively the center and variate, $D^j$ denotes the $j$-th differential, $(y-x)^{\otimes j}$ denotes the $j$-fold tensor product, and $O(\|y-x\|^{k+1})$ denotes the class of functions that vanish at least as fast as a degree-$(k+1)$ polynomial $M\|y-x\|^{k+1}$ as $y \to x$ for some $M > 0$. The $k$-th order jet operator of a function $f$ maps vectors to equivalence classes of degree-$k$ polynomial functions (we denote the resulting quotient space by $P^k$ in the equation below, details in the appendix) as follows:

$$J^k f : \mathbb{R}^d \to P^k \qquad J^k f(x) = f(x) + \sum_{j=1}^{k} (j!)^{-1} D^j f(x). \tag{7}$$

Evaluating the jet at a variate $y \in \mathbb{R}^d$ yields the truncated Taylor expansion $J^k f(x)(y) \in \mathbb{R}^d$, that is, eq. (6) without the "$O$" term. The main advantage of working with jets rather than Taylor expansions is that we can work directly with functions rather than vectors. We will make extensive use of the following lemma. Its proof, alongside further details about jets, is in appendix A.

**Lemma 1** (Convex combinations of jets). *Let $f \in C^\infty(\mathbb{R}^d, \mathbb{R}^d)$, $k \in \mathbb{N}$, and $\mathcal{C} = \{x_i\}_{i\in[N]}$ be a set of centers, for some $N \in \mathbb{N}^+$. Then,*

$$J^k f\left(\sum_{i=1}^{N} x_i\right) = \sum_{i=1}^{N} w_i J^k f(x_i) + O(r(w, \mathcal{C})^{k+1}) \quad \text{for any } w \in \triangle^{N+1},$$

*where $r(w, \mathcal{C}) = \max_i \{w_i \|x_i - \sum_j x_j\|\}$. We call any vector $w$ in the simplex a jet weight.*

**Remark 1** (Jet centers and variates as functions). *We will often want to trace the computation of a jet back to the input space $\mathcal{Z}$. In such cases, we interpret the jet centers $x$'s and the variates $y$'s as functions of the original network input $z \in \mathcal{Z}$ onto $\mathbb{R}^d$ or $\mathcal{Y}$. Thus, we have that $J^k f(x)(y) : \mathcal{Z} \to \mathbb{R}^d$ (or $\mathcal{Y}$) which evaluates as follows: $J^k f(x)(y)(z) = J^k f(x(z))(y(z))$.*

**Exponential expansion of a two-blocks network** Before introducing the main algorithm, we start with a minimal example of an expansion of a network with two residual blocks into four input-to-output paths. The network, represented in fig. 1 (a-I) and (a-II), is given by:

$$q = v \circ h; \quad h_2 = \beta_2 \circ \beta_1 \circ \eta = \eta + \gamma_1 \circ \eta + \gamma_2 \circ (\eta + \gamma_1 \circ \eta) \tag{8}$$

The final residual stream $h_2$ is a sum of three terms (input-to-hidden-space functions). In a transformer network, $\gamma_1$ could represent a self-attention block and $\gamma_2$ an MLP block – typically both transformations being input-normalized. Critically, the last term $\gamma_2 \circ (\eta + \gamma_1 \circ \eta)$ does not allow us to directly single out contributions that involve $\gamma_2$ and $\eta$ *or* $\gamma_1 \circ \eta$ alone. To recover such paths, we can jet-expand $\beta_2$ and apply lemma 1 choosing as centers $x_\emptyset = \eta$ and $x_{\{1\}} = \gamma_1 \circ \eta$, obtaining:

$$J^k \beta_2(x_\emptyset + x_{\{1\}}) = w_1 J^k \beta_2(x_\emptyset) + w_2 J^k \beta_2(x_{\{1\}}) + O(r^{k+1})$$
$$= x_\emptyset + x_{\{1\}} + w_1 J^k \gamma_2(x_\emptyset) + w_2 J^k \gamma_2(x_{\{1\}}) + O(r^{k+1}_{\beta_2}), \tag{9}$$

where the last equality holds for $k \geq 1$. [2] This operation is represented in fig. 1 (b). These terms still do *not* yield input-to-output paths, as in general $\gamma_3 \neq \mathrm{id}$ (in transformer architecture this is typically a normalization operation, e.g. layer norm). We can again proceed with a jet expansion, this time of the decoding module $\upsilon = U\,\gamma_3$. Continuing with our example, we apply lemma 1 using as centers the outputs of the previous expansion, namely $x_\emptyset$, $x_{\{1\}}$, $x_{\{2\}} = w_1 \mathrm{J}^k \gamma_2(x_\emptyset)$ and $x_{\{1,2\}} = w_2 \mathrm{J}^k \gamma_2(x_{\{1\}})$, obtaining

$$\mathrm{J}^k \upsilon(x_\emptyset + x_{\{1\}} + x_{\{2\}} + x_{\{1,2\}}) = \sum_{S \in 2^{[2]}} \omega_1 U \, \mathrm{J}^k \gamma_3(x_S) + O(r_\upsilon^{k+1}) \qquad (10)$$

where $\omega \in \Delta^3$ is a vector of jet weights. With this operation, represented by fig. 1 (c), we have obtained four input-to-output paths, mimicking the exponential rewriting of the linear case; cf. Equation (5). For instance, the zeroth order ($k = 0$) path that passes through the second non-linearity only, skipping the first, is given by the function $z \in \mathcal{Z} \to \omega_3 U \gamma_3(w_1 \gamma_2(\eta(z))) \in \mathcal{Y}$. This example demonstrates the key principles of our approach: recursive expansion of the computational graph using jets, and the use of convex combinations of jets to isolate specific paths. However, for deeper networks with many blocks, manually expanding each layer becomes impractical. To address this, we generalize this process into an algorithmic framework, which we develop next.

**jet-expand algorithm** Algorithm 1 presents the key operation of the framework. The algorithm applies lemma 1 to a residual transformation or to the decoding non-linearity for a given (user-defined) set of centers $\mathcal{C}$. It yields a set of expanded polynomial terms $\xi$, which can be seen as a set-valued function $\xi : \mathcal{Z} \times \triangle^{N-1} \to \mathcal{E}$, where $\mathcal{E}$ is an appropriate power set of functions, and a non-linear remainder $\delta : \mathcal{Z} \times \triangle^{N-1} \to \mathbb{R}^d$. The remainder encompasses both the residuals stemming from eq. (6) and lemma 1. As we showed above, centers can be the outputs of

---

**Algorithm 1** $\texttt{jet\_expand}(q, l, \mathcal{C}, k)$

**Require:** Residual net $q$, block index $l \in [L]$; jet centers $\mathcal{C} = \{x_i\}_{i \in [N]}$; order $k \in \mathbb{N}$;
**Ensure:** $\xi$ is a set of (partial) jet paths with weights $w \in \triangle^{N-1}$ and $\delta$ is a reminder.
1: $\xi \leftarrow \{w_i \mathrm{J}^k \gamma_{l+1}(x_i)\}_{i \in [N]}$
2: **if** $l < L$ **then**
3: $\quad \xi \leftarrow \xi \cup \{w_i \mathrm{J}^k \mathrm{id}(x_i)\}_{i \in [N]}$
4: $\quad \delta \leftarrow h_{l+1} - \sum_{e \in \xi} e$
5: **else** $\delta \leftarrow \gamma_{L+1} \circ h_L - \sum_{e \in \xi} e$

---

previous expansions, enabling the propagation of the expansion through the entire network and effectively 'unrolling' the computation graph into distinct paths. Importantly, once we apply the algorithm for $l = L$ we obtain a way to *rewrite the computational graph* of $q$ as a sum of expanded terms (input-to-output paths), which we call *expansion*, and a non-linear remainder. Indeed, if $(\xi_L, \delta_L) = \texttt{jet\_expand}(q, L, \mathcal{C}, k)$ for some $\mathcal{C}$ and $k$, the following class of functional equivalences holds:

$$q = \sum_{e \in \xi_L} U e(\cdot, w) + \delta_L(\cdot, w) \qquad \text{for } w \in \triangle^{N-1}. \qquad (11)$$

**Runtime** The runtime of algorithm 1 is negligible as it operates on the original computational graph. Evaluating $\xi$ (and $\delta$) at any $z \in \mathcal{Z}$ requires computing $k$th-order jets with a complexity of $O(|\mathcal{C}|(F + kB))$, where $F$ and $B$ are the costs of forward and backward evaluations of $q$. In practice, higher-order jets can be computed efficiently using stored computation (Griewank & Walther, 2008; Bettencourt et al., 2019). Specifically, $k$-th order derivatives can be computed using recurrence $D^k f(x)(y - x) = \texttt{jvp}(D^{k-1}f, x, y - x)$, where $\texttt{jvp}$ computes the Jacobian-vector product and it is available in most mainstream automatic differentiation frameworks like Pytorch. Appendix B reports an example of runtime scaling with the jet order $k$ in our implementation.

**Remark 2** (Jet weights optimization). *Jet weights $w$ can be fixed, e.g. $w_i = 1/N$ or optimized to minimize the remainder at any given $z$, such as after projection into the logit space. This optimization can be done efficiently as $\|U \delta_L(z, w)\|^2 = \|\gamma_L(h_L(z)) - \sum_{e \in \xi_L} e(z, w)\|_{U^\top U}^2$, which amounts to the squared distance between the expansion and the original residual stream in the representation space $\mathbb{R}^d$ with the metric induced by the unembedding matrix. In our jet lens experiments in section 5.2, we optimized jet weights with gradient descent.*

**Remark 3** (Non-vanishing remainders). *In general, we cannot expect reminders to vanish (as $k$ grows). Indeed, even if the convergence radius of the Taylor series is infinite, the arguments of residuals introduced by applications of Lemma 1 do not vanish. If $q$ is a linear residual network,*

---

[2]For $k = 0$ the weights apply also to the center terms since $\mathrm{J}^0 \mathrm{id}(x_{\{1\}} + x_{\{2\}}) = w_1 x_{\{1\}} + w_2 x_{\{2\}} + O(r^1)$.

*however, $\delta = 0$ for $k \geq 1$, showing that Algorithm 1 recovers (after reorganizing terms) the rewrite of Equation (5) for every choice of $w$. [3] Hence, in light of Equation (11), jet expansions should be seen as ways to rewrite computational graphs rather than function approximations – how close the expansions are to the model output depends on the specific choice of centers and order. In experiments we show however how $\delta$'s can be small and the cosine similarity between expansion and original network logits can be close to 1; see Figure 3.*

## 4 NOTABLE EXPANSIONS AND THEIR IMPLICATIONS

We introduce some particular expansions as application of the introduced `jet_expand` algorithm, setting the stage for the numerical case studies of the next section.

**(Super)exponential expansion.** Algorithm 2 generalizes the exponential expansion we performed onto the two-blocks network in section 3, using uniform jet weights. One can interpret the algorithm as performing a "maximal" expansion (when remaining at the grain of the blocks) which yields $2^L$ input-to-output paths. In fact, for $k \geq 1$, we can further isolate each degree of the expanded terms into separate input-to-output paths that highlight interactions among various blocks. This further refinement,

---

**Algorithm 2** `exp_jet_expansion`$(q, k)$

---

**Require:** Residual network $q$; order $k \in \mathbb{N}$;
**Ensure:** $\xi$ is a set of equally weighted input-to-output jet paths, $|\xi| = 2^L$, and $\delta$ is a reminder.
1: $\xi \leftarrow \{\eta, \gamma_1 \circ \eta\}$
2: **for** $l \in [L]$ **do**
3: $\quad (\xi, \delta) \leftarrow$ `jet_expand`$(q, l, \xi, k)$
4: $\quad \xi \leftarrow \{e(\cdot, 1/|\xi|)\}_{e \in \xi}$

---

which we will focus on in future work, may suggests that residual networks may in fact behave as super-exponential ensembles of (shallower) functions.

**Jet lenses and the logit lens.** The logit lens (nostalgebraist, 2021b; Geva et al., 2021; 2022; Merullo et al., 2023; Belrose et al., 2023) is an interpretability method that consists in applying the decoder to intermediate representations as follows:

$$\text{LogitLens}_l(z) = U\gamma(h_l(z)) = \text{J}^0 \upsilon(h_l(z))(h_L(z)).$$

The logit lens, aimed at highlighting the iterative refinement of the prediction across blocks, is related to early exiting (or early decoding) in the context of conditional computation (see e.g. Panda et al., 2016; Elbayad et al., 2020; Geva et al., 2022). It is immediate to verify that $\text{LogitLens}_l$ is equivalent to the expansion yielded by `jet_expand`$(q, L, \{h_l\}, 0)$. This suggests two generalizations, which we dub *iterative* and *joint* jet lenses, respectively. The iterative jet lens is a direct extension of the logit lenses that allows for higher order jets: `jet_expand`$(q, L, \{h_l\}, k)$. The joint jet lenses are expansions obtained through `jet_expand`$(q, L, \{\gamma_l \circ h_{l-1}\}_{l \in [L]}, k)$ that are aimed at highlighting the residual contributions of each block non-linearity, rather than the iterative refinement of the residual stream.

**Jet bi-grams and skip-$n$-grams statistics.** We consider transformer-based large language models with alternating self-attentions and MLPs, which are particular instances of residual nets. [4] Our framework allows us to directly extract $n$-gram statistics from an existing LLM without any probing datasets. Concretely, we can systematically evaluate relevant jet paths (for small $n$'s) on the entire input space, usually the vocabulary and its Cartesian products, independently from individual contexts. For example, bi-grams statistics related to $\mathbb{P}_q(z_2|z_1, \dots)$ can be computed by evaluating bi-gram paths, which we can obtain by expanding the LLM with Algorithm 2 and filtering out all paths that involve self-attention modules. Specifically in our case studies (Section 5.1), we focus on encoding-decoding bi-gram path, obtainable via expanding the LLM with `jet_expand`$(q, L, \{\eta\}, k = 0)$, and the bi-gram paths involving up to one MLP module, which can also be obtained via applying Algorithm 1 twice. We can obtain skip-$n$-gram statistics relating to $\mathbb{P}_q(z_n|z_{n-1}, \dots, z_{n-2}, \dots, z_1, \dots)$, where dots indicate any number of interceding tokens, by evaluating jet paths with self-attentions (the fewer self-attentions, the lower the $n$) and isolated single query-key products. Such jet $n$-gram statistics offer a *data-free* tool to sketch LLMs via casting them into (symbolic) $n$-gram databases. Thus they allows us to perform symbolic model diffing between *any* two models that share a common vocabulary, as opposed to take differences in the parameter space, harder to interpret and only possible for same-architecture models.

---

[3]Other special cases include expansions where each center set is a singleton and the convergence radius of the expanded non-linearities is infinite.

[4]We disregard positional embeddings for simplicity and leave their study to future work.

| | new | _simple | _neural | _architecture | , | _the | _Trans | former |
|---|---|---|---|---|---|---|---|---|
| Block 1 (7.36%) | , (3.40%) | ton (8.06%) | _network (8.57%) | _for (8.22%) | _which (7.51%) | _first (7.30%) | former (7.43%) | , (8.36%) |
| Block 2 (4.83%) | - (2.39%) | _ (5.23%) | _network (6.91%) | _for (4.98%) | _which (4.60%) | _neural (4.77%) | former (5.09%) | , (4.68%) |
| Block 4 (7.81%) | _impover (1.62%) | _unpop (1.29%) | _impover (1.31%) | _impover (1.28%) | _impover (1.25%) | _Neural (1.22%) | former (1.20%) | _Networks (1.32%) |
| Block 24 (6.02%) | , (5.74%) | _infographic (8.48%) | _network (8.76%) | _unve (8.45%) | _unve (7.67%) | _Neural (7.51%) | former (7.39%) | _model (8.45%) |
| Block 30 (6.24%) | âĢ!" (5.29%) | _ (1.31%) | _network (1.30%) | _for (1.29%) | _which (1.29%) | _neural (1.26%) | former (1.25%) | Ãı (1.31%) |
| Block 31 (7.76%) | !!" (5.33%) | _ (1.33%) | _network (1.31%) | _for (1.29%) | _the (1.26%) | _Conv (1.23%) | former (1.23%) | , (1.32%) |
| Block 32 (7.84%) | âĢ¦" (3.56%) | !?" (1.37%) | _network (1.36%) | , (1.33%) | _and (1.28%) | _neural (1.24%) | former (1.25%) | _model (1.32%) |

| | new | _simple | _neural | _architecture | , | _the | _Trans | former |
|---|---|---|---|---|---|---|---|---|
| Logits | _ | _ | _network | _for | _which | _neural | former | , |
| Expan. (0.993) | _ | _ | _network | _for | _which | _neural | former | , |

Figure 2: Example of a joint jet lens on *GPT-Neo* 2.7B with $k = 1$, visualizing the seven blocks with highest average jet weights after optimization. Each table cell indicates the most likely token of the jet path related to each block non-linearity. Optimized jet weight are in brackets. We used a diverging blue-to-red color map tracking logit scores, centered around zero. The bottom table shows the model logits and the expansion logits, with cosine similarity in brackets $0.993$; in this case, all top-1 tokens perfectly coincide.

## 5 INTERPRETING LLMS WITH JET EXPANSIONS

Our framework provides users with freedom in terms of choosing the computational paths they wish to focus on. Jet expansions support studies across various levels, including model-level global analysis (jet $n$-grams), component-level analysis (jet paths), and example-level analysis (jet lens).

### 5.1 SETUP

We experiment with several popular open-sourced large language models families: *GPT-2* (Radford et al., 2019), *GPT-Neo* Black et al. (2021), *Llama* (Touvron et al., 2023a;b; Rozière et al., 2024) and *OLMo* (Groeneveld et al., 2024), showcasing the generality of the algorithm. Our main experiments run on $128$ CPU servers with $1$ TB memory, while jet lens experiment run on a single laptop. The experiments on jet lenses uses higher-order jet. We optimize jet weights of joint jet lenses with gradient descent, minimizing the loss introduced in remark 2. In the rest of the experiments, we use zeroth order jet bi-grams (from the paths that go through MLPs and the direct embedding-unembedding paths) and tri-grams (from the paths that pass through the corresponding attention heads). Each path $e_l : \mathcal{Z} \to \mathcal{Y}$ is obtained by applying algorithm 1 twice (expect for the embedding-unembedding path, which requires only one call): if $\gamma_l$ is the non-linearity of interest (either an MLP or a self-attention head), we first call $\hat{e}_l, \hat{\delta} = \texttt{jet\_expand}(q, l, \{\eta\}, 0)$ and then call $\tilde{e}_l, \delta = \texttt{jet\_expand}(q, L, \{\hat{e}, (\cdot, 1)\}, 0)$, finally setting $e_l = \tilde{e}(\cdot, 1)$. [5] We further detail algorithmic procedures in appendix C.

We define some metrics used in our empirical study. 1) $\Delta$ **Logit after Intervention.** We measure the logit for an $n$-gram before and after applying an intervention (e.g., removing an attention head) and compute the change at the last position. 2) **One-to-One and Many-to-Many Bi-grams.** One-to-one bi-grams are unimodal, concentrating probability on a single token (e.g., $z_1 = \&$, $z_2 = $ amp). Many-to-many bi-grams have multi-modal distributions, where multiple tokens can follow $z_1$ or precede $z_2$ (e.g., $z_1 = $ make, $z_2 = $ sure). 3) **Total Mass of Key Bi-grams.** The total mass metric measures the cumulative probability of the top 1K bigrams, weighted by an empirical unigram distribution. Formally, it is Total Mass $= \sum_{(z_1, z_2) \in \text{Top-1K}} \mathbb{P}_{e_t}(z_2|z_1)\mathbb{P}_{\mathcal{D}}(z_1)$, where $e_t$ is the embedding-unembedding path at step $t$, $(z_1, z_2)$ are the bigrams, $\mathbb{P}_{e_t}(z_2|z_1)$ is the model probability of $z_2$ given $z_1$, $\mathbb{P}_{\mathcal{D}}(z_1)$ is the unigram probability of $z_1$ from the empirical distribution. This metric evaluates how well the model assigns "correct" probability mass to bigrams, considering the unigram probability of $z_1$, and reflects alignment with the empirical distribution during pretraining.

### 5.2 ANALYZING LLM INNER WORKING

LLMs are notorious for their lack of interpretability due to their inherent model complexity and size, made worse by the usual opaque training process and unknown training data. Understanding their inner working contributes to calibrating trust for users to use them appropriately. We showcase how jet expansion along user-selected computational paths (jet paths) can help us discover and locate learned associations akin to studies in mechanistic interpretability Templeton et al. (2024).

**Jet lenses.** We use jet lenses introduced in Section 4 to analyze LLM's mechanism when processing individual examples. Figure 2 visualize a joint jet lens for GPT-Neo 2.7B (Black et al., 2021)

---

[5]With a small abuse of notation, we identify singleton sets with their single member.

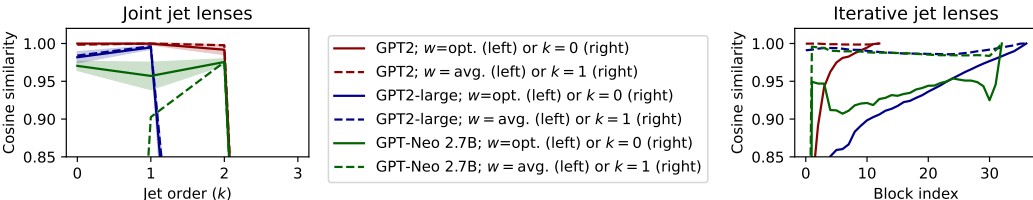

Figure 3: Plots of average cosine similarities between model logits and jet lenses logits. (Left) jet lens of the joint variant (**left**) and jet lens of the iterative variant (**right**). In the right plot, the solid lines of all colors correspond to the LogitLens ($k = 0$), and dashed lines to the iterative jet lens for $k = 1$.

(other examples can be found in Appendix H). Here, a block contains one self-attention and one MLP module. All table cells depict top-1 tokens for the corresponding path, following conventions from prior work (Belrose et al., 2023). We observe that the joint jet lens captures the synergy among different blocks, as the model prediction is decomposed into several jet paths. Our preliminary analysis supports recent work on super-position (Elhage et al., 2022) and neuron polysemy (Bricken et al., 2023), suggesting that interactions among components may have ensemble effects, which can broadly vary across model families. In this sense, the jet lenses with $k > 0$ may serve as tools to systematically discover such synergic behaviors. We also find that higher-orders ($k > 0$) help iterative lenses deliver more meaningful interpretations than the logit lens ($k = 0$) for *GPT-Neo*-2.7B (see Figures 7 to 9). This is potentially due to their capability to trace indirect impacts of early layers on the final logits, which were otherwise missing under logit lens. Our findings are consistent with nostalgebraist (2021a); Cancedda (2024) where naive implementations of logit lens are shown to fail on *GPT-Neo* model family. Figure 3 present cosine similarities (against the original model logits) of joint and iterative jet lenses for various *GPT* models and jet orders, averaged over 100 example sentences. The similarities are high and close to 1 for various $k$'s, showing however different behavior across model families and sizes. In particular, the right plot compares the similarities of the logits obtained through iterative jet lenses for $k = 0$ (solid, line, the same as LogitLens) and for $k = 1$ (dashed lines), indicating an higher correlation of the latter with model outputs, potentially providing more faithful interpretations.

**Jet paths of individual components.** By examining the representative jet bi-grams that are captured by each MLP path, we find some MLPs that perform special linguistic functions. For example, in *OLMo*-7B, the jet path which passes through the 3rd MLP promotes the addition of the "`-ing`" suffixes to the current token. Similar MLPs with certain linguistic functions are listed in Table 1. Note that the relationship between functions and components are not necessarily one-to-one mappings. Particularly we find that the paths through multiple MLPs might work together to complete one linguistic function e.g. MLP 6 and MLP 18 in *Llama*-2-7B can add "`-ing`" suffix. One MLP might also do multiple linguistic jobs e.g. MLP 1 in OLMo 7B adding "`-ly`" and "`-_else`" suffixes. This echos work on circuit discovery (Conmy et al., 2023; Ferrando & Voita, 2024) and superposition (Elhage et al., 2022), where the role of each component cannot be easily dissected and multiple components collaborate to fulfill a function. Table 2 reports a role identification study on attention heads in the first self-attention of *OLMo*-7B using jet tri-grams. Specifically, we find heads associated with math and programming, e.g. head 1 on Math/Latex; heads promoting digits and dash composition into dates, e.g. head 25; and heads constituting phrase templates, e.g. head 15 managing a "for $x$ purposes", where $x$ is a placeholder. To verify the roles we revealed, we further perform preliminary intervention experiments where we ablate MLPs or attention heads and compute variations in model logits. After the interventions, the logits drop consistently in all cases, suggesting our jet $n$-grams indeed can help identify certain roles for selected components. Varying impact on logit differences is likely due to overdetermination (Mueller, 2024) and our partial selection of jet paths (e.g. for tri-grams we only selected encoding-attention-decoding paths, excluding any MLP).

## 5.3 ANALYZING PRETRAINING DYNAMICS

Pretraining an LLM is usually extremely resource intensive. Therefore it is crucial to monitor the progress of a pretraining run to prevent wasting of time and compute. In this section, we show how jet bi-grams can serve as an effective signaling tool to trace the pretraining dynamics, providing insights about the model's maturity. Such signals are especially useful to understand what happens

Table 1: MLPs in *OLMo-7B* and *Llama-2-7B* performing certain linguistic functions based on jet bi-grams extracted from the corresponding jet paths.

| | OLMo-7B | | | | | Llama-2-7B | | | |
|---|---|---|---|---|---|---|---|---|---|
| MLP Index | 1 | 3 | 9 | 17 | 19 | 6 | 7 | 18 | 19 |
| Role | -ly, -else | -ing | -'t | -than | -s | -ing | -es | -ing,-ity | -ly |
| Δ logit after intervention | −4.19, −3.35 | −0.58 | −9.73 | −4.26 | −7.42 | −14.61 | −3.55 | −9.69, −11.93 | −9.14 |

Table 2: Several attention heads in the first residual block of *OLMo-7B* and their roles identified with jet tri-grams extracted from corresponding jet paths. We also include an example tri-gram captured by each head.

| Head Index | 2 | 16 | 26 | 30 |
|---|---|---|---|---|
| Role | Math/LaTeX | "for ... purposes" | date composition | "into account/consideration ..." |
| Example 3-gram | (_Lemma, _let, _s) | (_for, _use, _purposes) | (20, 23, _-) | (_into, _account, _possible) |
| Δlogit after intervention | −0.1570 | −0.0019 | −0.0093 | −0.0001 |

with the model when the pretraining loss shows marginal improvements and fails to reflect the changes inside the model.

**Identifying the top bi-grams.** To assess the model's progression, we extracted jet bi-grams from *OLMo-7B* model checkpoints across 555K pretraining steps. Table 4 presents a summary of the top 10 jet bi-grams at different stages of training. Due to space reason, we only show the top 10 jet bi-grams every 100K steps. Initially, the network exhibits nonsensical jet bi-grams, such as "ICUirling". As training advances, it gradually learns more meaningful combinations, like "at least". This process of acquiring sensible bi-grams stabilizes around step 200K, indicating that the model is reaching a level of maturity where the top 10 bi-grams capture common meaning.

**Learning schemes for different bi-grams.** To understand if there are any differences between the learning schemes of different bi-grams, we can trace the progression of the jet bi-gram scores for selected bi-grams. Figure 4 provides a visual comparison of how different bi-grams are promoted or suppressed during the pretraining process. The different slopes and levels of the lines indicate varying rates of learning for the respective bi-grams. We observe that, the model first acquires random bi-grams due to random parameter initialization. These random bi-grams, like "ICUirling" and "VENT thanks", are quickly suppressed in the early steps and never regain high scores. In contrast, one-to-many bi-grams like "at least" are first promoted to very high scores but then get suppressed perhaps due to the model seeing more of the scope of the token "at". One-to-one bi-grams like "&amp" (HTML code) are gradually promoted and stabilize. Many-to-many bi-grams like "make sure" takes the most time to learn and the scores are still increasing even at the end of pretraining. Our findings suggest that the training process effectively promotes certain "good" bi-grams, but at different paces, where they might be suppressed later depending on their occurrences and linguistic nature. These insights could inform future training strategies, such as targeted training on more relevant bi-grams or adjusting the training data to improve the pretraining speed.

### 5.4 ANALYZING FINE-TUNING EFFECT

Fine-tuning is an important phase where the raw pretrained LLMs are guided to perform particular tasks. We would like to understand how the model inner knowledge changes during fine-tuning processes. While parameter diffing can be a straightforward solution, jet n-grams provides an alternative approach, where the diffs are human readable and directly reflect the change of knowledge retained by the LLMs. Such insights would allow us to better decide the mixture of data for fine-tuning, and the number of steps for fine-tuning, which are currently a mix of heuristics and trial-and-error.

**Code fine-tuning promotes coding-relevant bi-grams.** We analyze the changes due to code fine-tuning via *diffing* jet bi-grams extracted from *Llama-2-7B* and its fine-tuned versions, *Codellama-7B* and *Codellama-Python-7B*. As highlighted in Table 5 with orange coloring, the jet bi-gram diff reveals coding-relevant keywords, suggesting jet bi-gram can be a tool for verifying if the fine-tuning is effective.

**Does RLHF fine-tuning remove toxicity?** We compare the original pretrained model, *Llama-2-7B*, with its RLHF version, *Llama-2-7B-Chat*. RLHF alignment (Bai et al., 2022) is widely believed to detoxify LLMs, as indicated by the *ToxiGen* scores (Hartvigsen et al., 2022). However, it remains

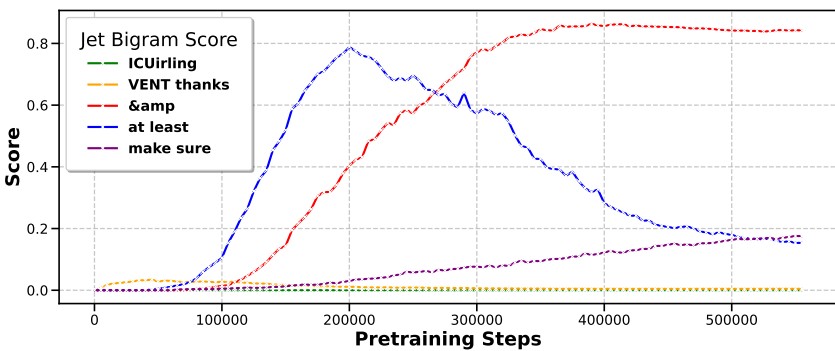

Figure 4: Visualization of *OLMo-7B*'s promotion and suppression dynamics of jet bi-grams scores.

Table 3: Toxicity indexes for *Llama-2-7B* and *Llama-2-7B-chat* using different methods: *ToxiGen*, jet bi-grams, and *RealToxicityPrompts* challenge prompting. Higher numbers indicate higher toxicity scores on the corresponding benchmarks and higher toxic knowledge possession for jet bi-grams.

| | ToxiGen Score | Jet Bi-grams | RTP Challenging Prompts | | | |
|---|---|---|---|---|---|---|
| | Hartvigsen et al. (2022) | Mass of "toxic" bi-grams | No | Very mild | Medium | Hard |
| *Llama*-2-7B | 21.25 | 0.03445 | 38% | 49% | 64% | 88% |
| *Llama*-2-7B-chat | 0.0 | 0.03377 | 23% | 35% | 64% | 84% |

easy to prompt LLMs to bypass this alignment and produce toxic content. In table 3, we demonstrate this with dataset-based toxicity scores on a subset of challenging prompts in the *RealToxicityPrompts* (RTP) dataset (Gehman et al., 2020): the gap in toxicity potential between the two models *narrows* as we prepend to RTP prompts increasingly "explicit" (short) context. Specifically, for hard context, *Llama-2-7B-Chat* shows an 84% probability of producing toxic content, close to that of *Llama-2-7B*. This suggests that the RLHF model is not completely detoxified but rather hides the toxicity knowledge from the "surface", which however can be easily triggered by specific contexts. To quantify the toxicity knowledge embedded in these models, we use jet bi-gram probability scores and calculate the cumulative conditional probability mass for a set of "toxic" bi-grams, which are combinations of tokens associated with toxic meanings from a predefined list of keywords. Interestingly, we observe a small change in mass from 0.03445 to 0.03377 after RLHF. Thus, although the *ToxiGen* score may suggest that the model has been effectively detoxified, the jet bi-gram mass reflects retention of toxic knowledge after RLHF, aligning with the scores obtained by introducing medium or hard explicit context and computing a toxicity score (via a second scorer model, (Hanu & Unitary team, 2020)) on *RealToxicityPrompts* dataset (Gehman et al., 2020). This showcases a potential application of jet bi-grams in constructing *data-free* indices that reveal embedded knowledge, offering complimentary views beyond traditional data-driven benchmark evaluations.

## 6 RELATED WORK

**Interpreting transformers.** There has been much effort in interpreting the inner computations of transformer models. In particular, *mechanistic interpretability* Ferrando et al. (2024), focuses on reverse-engineering such computations by identifying, clustering and labelling model behavior (Shah et al., 2024; Meng et al., 2022; Bricken et al., 2023) in human understandable terms and attributing them with certain model components, e.g., MLPs Geva et al. (2021; 2022), or typical "circuits" (Conmy et al., 2023; Ferrando & Voita, 2024). Authors discussed limitations of currents approaches to MI. For example, Templeton et al. (2024) found it generally hard to conclude neuron-level intepretabilities, compared with feature representations; while Bolukbasi et al. (2021); Goldowsky-Dill et al. (2023) points out that conclusions drawn are generally limited to the chosen data distribution. On a high level, allowing taking any portion of compute out of the original transformer, jet expansions abstract and generalize previous characterizations on the computational paths (Veit et al., 2016; Elhage et al., 2021), where non-linear components with significant roles, e.g. layernorm and MLPs, are either ignored or over-simplified for the ease of analysis. Our approach also does not require extra datasets that are used for probe fitting in methods such as Belrose et al. (2023) nor sampling, as needed in (Conmy et al., 2023; Ferrando & Voita, 2024; Voita et al., 2024).

$n$-**gram models.** The early applications of $n$-gram models in languages dates back to (Shannon, 1948), where $n$-grams modeled the statistics of English. The $n$-gram based approaches have been an important baseline in language processing, e.g., general language modelling (Goodman, 2001) with applications like machine translation (Brants et al., 2007). There have been regained interests on combining $n$-gram with neural network model-based approaches (e.g. Liu et al., 2024). Several recent works have explored the relationships between LLMs and $n$-gram language models, such as analyzing the representational capacity of transformers to simulate $n$-gram LMs (Svete & Cotterell, 2024) and measuring agreement between LLM predictions and $n$-gram rulesets (Nguyen, 2024).

**Taylor expansion and jets** Taylor expansions are popular tools in analyzing learning behaviours (Jastrzebski et al., 2017), notably linearization ($k = 1$). For example, Belrose et al. (2024) applied Taylor expansion on the loss function to demonstrate the learning preference of neural network models. Xu et al. (2022) introduced a second-order Taylor expansion over the data distribution to interpret optimal features. The generalized jet notions was introduced in machine learning in the context automatic differentiation tools by Bettencourt et al. (2019), and is an experimental feature in Jax (Bradbury et al., 2018), but has been studied before (see e.g. Griewank & Walther, 2008).

# 7 CONCLUSION AND DISCUSSION

We introduced *jet expansion*, a novel framework for expanding the computational graphs of neural networks. The method, which we specialize in this paper to deep residual nets, can be used to disentangle contributions of user-selected computational paths from the overall graph. Complementary to other dataset-dependent methods in MI, our method enables various dataset-free global interpretability studies, such as mapping computation to linguistic roles. We have validated jet expansions in terms of cosine similarity against model outputs and through interventional experiments (section 5.2). We applied our data-free method to transformer LMs, showing how we can sketch the original model with input-output probability databases, extracting LM bi-and-tri-gram statistics.

**Limitations.** Although rooted in Taylor series theory, expansions obtained via our frameworks do not (seek to) approximate the input function in any strict sense. Rather, our framework is amed at facilitating interpretation of model behavior: we can use jet expansion *to rewrite* an input computational graph as a sum of "interpretable" polynomial terms and a (computable) remainder. How large is a reminder and how expansions align with model outputs remains at the moment an empirical question, implying that the jet order and weight optimization routines should generally be considered as hyperparameters of the method. Furthermore, expansions are not unique (but higher order expansions "contain" lower order one). We leave a deeper investigation of these aspects to future work. From a runtime standpoint, we note that even though graph manipulation is almost immediate, systematic evaluation of jet paths may be time consuming (especially for $k \gg 0$ and when optimizing jet weights). If the input space is large, one may need to resort to sub-sampling or search heuristics. Finally, we limited our study of $n$-gram expansions of LMs to bi-and-tri-grams, unearthing compelling behaviors. This leaves the study of longer-context expansions to future work.

**Implications and future work.** Our work opens up several research directions. From a theoretical standpoint, we will extend the expansion procedure to cover finer granularities, e.g. at neuron (subspace) levels; incorporate established attribution methods such as the Shapley value (Shapley et al., 1953), including recent extensions to deal with probabilistic models (Franceschi et al., 2024); develop concepts of (approximate) equivalence classes over models leveraging the jet spaces, which, in turn, may further ground the model diffing procedure sketched in our case studies. Furthermore, we will take inspiration from the numerous tools in linear algebra to provide further depth into the analysis, deepening the link to linear residual structures and establishing relations with Markov chains and hidden Markov models, recently employed e.g. by Zhang et al. (2023) for constrained (structured) decoding. We plan to investigate the implication of the super-exponential number of paths in the residual networks depth unearthed by algorithm 2. From an applications standpoint, besides studying jet $n$-grams for $n > 3$, we envision several fruitful applications in safety and transparency, such as developing "search features" to systematically detect unwanted associations, or leaked private content. Although our experiments are primarily observational, we speculate that `jet_expand` may also become an useful tool to guide interventions, supplementing other techniques like causal tracing (Meng et al., 2022) and path patching (Goldowsky-Dill et al., 2023).

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

# A    ADDITIONAL DETAILS ON JETS

A jet of a function represents an equivalence class. We thus can perform algebraic operations among functional equivalence classes using jet algebra stated below.

**Proposition 1** (Jet algebra). *Let $f, g \in C^\infty(\mathbb{R}^d, \mathbb{R}^d)$ and $k \in \mathbb{N}^+$. Then,*

*(i)* $\mathrm{J}^k(af + bg)(x) = a\,\mathrm{J}^k(f)(x) + b\,\mathrm{J}^k(g)(x)$, *for $a, b \in \mathbb{R}$ (linearity);*

*(ii)* $\mathrm{J}^k f(x) \circ g \in \mathrm{J}^k f(x)$ *and* $\mathrm{J}^k f(x) \circ g(y) = \mathrm{J}^k f(x)(g(y))$ *(jet after endomorphisms);*

*(iii)* $g \circ \mathrm{J}^k f(x) = \{g \circ u \,:\, u \in \mathrm{J}^k f(x)\}$ *(endomorphism after jet);*

*(iv)* $\mathrm{J}^k(f \circ g)(x) = \mathrm{J}^k f(g(x)) \circ \mathrm{J}^k g(x)$ *(composition of jets);*

Properties *(i)-(iii)* follow directly from the definition; *(iv)* is a consequence of the chain rule and truncation.

**Proof of Lemma 1**    Take $y \in \mathbb{R}^d$, $N \geq 1$, $x_i \in \mathbb{R}^d$ for $i \in [N]$, $w \in \triangle^{N-1}$ and an order $k \geq 0$. Since $w$ belongs to the simplex $\triangle^{N-1}$, we have $\sum_{i=1}^N w_i = 1$. Multiplying $f(y)$ on both hands, we obtain

$$f(y) = \sum_{i=1}^N w_i f(y) = \sum_{i=1}^N w_i \left[ f(x_i) + \sum_{s=1}^k \mathrm{D}^s f(x_i)(y - x_i)^{\otimes s} + O(\|y - x_i\|^{k+1}) \right]$$

$$= \sum_{i=1}^N w_i \mathrm{J}^k f(x_i)(y) + O(w_i\|y - x_i\|^{k+1}),$$

by applying eq. (6) (Taylor expansion) and the definition of jet with each $x_i$ as the center. At the same time, we can expand $f(y)$ with $\sum_{i=1}^N x_i$ as the center

$$f(y) = \mathrm{J}^k f(\sum_{i=1}^N x_i)(y) + O(\|y - \sum x_i\|^{k+1}).$$

Now let us take $y = \sum_{i=1}^N x_i$ and observe that $O(\|y - \sum x_i\|^{k+1}) = 0$ and $O(w_i\|y - x_i\|^{k+1}) = O(w_i\|x_i - \sum_j x_j\|^{k+1})$. Finally we observe that the class of functions in the last $O$ are dominated by the class of function in $O(r^{k+1})$ where $r = \max_i\{w_i\|x_i - \sum_j x_j\|\}$ is the maximum remainder. This concludes the proof.

As a side note, jet weights would not need to form convex combinations, but rather linear combinations $\sum_i w_i = 1$. However, restricting to convex combinations has two major advantages:

- optimizing over a convex set guarantees the existence of maxima and minima (Weierstrass theorem) and uniqueness of minima if we are optimizing a strictly convex loss as in general is the case for expansions that only affect the decoder module.
- weights within the probability simplex have a clearer interpretation for interpretability purposes.

# B    ADDITIONAL DETAILS ON RUNTIME

We report in fig. 5 a plot of the runtime for evaluating expansions originating from the joint jet lenses of section 5.2 as a ratio of the input model evaluation (forward pass), for both the uniform and the optimized jet weights $w$ setup, for different jet orders $k$.

# C    ADDITIONAL DETAILS ON JET $n$-GRAMS

**General Concept of $n$-Gram Models**    The general concept of $n$-gram models linked to (transformer-based) LMs involves defining or constructing mappings that functionally depend only on $n-1$ input tokens (with the $n$-th token being the output token) to capture and describe the behaviour of the original LM. We are not the first to explore this idea; for instance Nguyen (2024) fits n-grams on the same dataset used to train the LM.

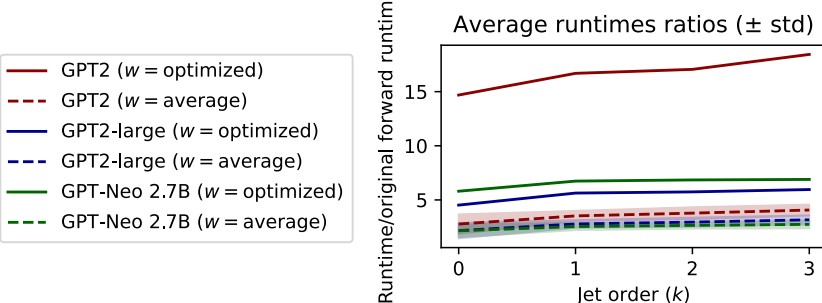

Figure 5: Empirical runtime of evaluations of jet expansions originating form the joint jet lenses as a ratio of the evaluation of the input model.

**Jet Expansions for In-Model $n$-Grams**  Jet expansions allow us to define $n$-grams statistics that are derived solely and directly from the model itself – producing in-model $n$-grams rather than in-data $n$-grams. This approach offers at least two significant advantages:

- **No Dataset Preparation:** It eliminates the need for dataset preparation to collect activation patterns when interpreting the model globally, thereby saving time and computational resources. This process can be conducted entirely on CPU, which is approximately 10 times cheaper per hour compared to GPUs in the current market.

- **Avoidance of Fitting Artifacts:** It avoids potential artifacts that could arise from the selection of external $n$-gram fitting methods.

We describe the detailed relationship between the bi-gram/tri-gram, which we used in our case studies, and the jet expansion as follows.

**Jet Bi-Grams**  Jet bi-grams are paths that do not pass through self-attention layers. In experiments, we focus on two types of bi-gram paths. a) the embedding-unembedding path that can be obtained as jet_expand$(q, L, \{\eta\}, 0)$. b) paths that pass through one MLP module, assuming MLPs are at odd block indices in the residual network architecture, the procedure to extract the path is:

$$
\begin{aligned}
\mathcal{C} =& \{\eta\} \\
&\text{for } l = 1, 3, \ldots, L-1: \\
&\quad \xi, \delta = \text{jet\_expand}(q, l, \{\eta\}, 0) \\
&\quad \mathcal{C} = \mathcal{C} \cup \{e(\cdot, 1)\}_{e \in \xi} \\
&\quad \xi, \delta = \text{jet\_expand}(q, L, \mathcal{C}, 0)
\end{aligned}
$$

This procedure results in a series of functions in $\xi$—one for each MLP layer—that depend only on the last input token. Applying softmax normalization to their logit output allows these functions to define (conditional) bi-grams. Similar constructions can be performed for paths through multiple MLPs. We will release code for these procedures and also provide equivalent algorithms that directly use transformer modules.

**Jet Tri-Grams**  Jet tri-grams involve paths that pass through at least one self-attention layer, with a need to isolate the contribution from the first token of the tri-gram. The procedure for extracting a 0-th order jet trigram path that passes through the $i$th self-attention layer (assuming it has one head and $\sigma_2$ is a function that extracts the last two tokens from a sequence of length at least 2) is as follows:

$$
\begin{aligned}
\text{Define} \quad & \sigma_2(z) = (z_{t-1}, z_t) \\
\text{Compute} \quad & \xi, \delta = \text{jet\_expand}(q, i, \{\eta \circ \sigma_2\}, 0) \\
\text{Compute} \quad & \xi, \delta = \text{jet\_expand}(q, L, \{e(\cdot, 1)\}_{e \in \xi}, 0)
\end{aligned}
$$

This procedure yields a map that depends only on two input tokens, isolating the contribution of the $i$th self-attention layer on pairs of tokens. Once softmax normalization is applied, this defines a tri-gram. The tri-gram could represent either a skip trigram or a contiguous trigram, depending on how positional information is encoded (e.g., absolute positional embeddings versus rotary embeddings).

## D  ADDITIONAL DETAILS ON THE EXPERIMENTAL METRICS

$\Delta$ **logit after intervention**    To compute $\Delta$ logits, we calculate the logits for the given $n$-gram both before and after applying the intervention, then determine the change in the logits. For example, consider the trigram (Lemma, let, s). We compute the logit of "s" conditioned on the input "Lemma let". The intervention involves removing the corresponding attention head (e.g., head 2). We then measure and report the change in the logit for "s" as a result of this intervention.

**One-to-one bi-grams like and Many-to-many bi-grams**    One-to-one bi-bigrams are (approximately) unimodal bi-grams that concentrate all mass on a single token: i.e. given $z_1$, $\mathbb{P}\_\mathcal{D}(z_2|z_1) \approx 1$ and given $z_2$, $\mathbb{P}_\mathcal{D}(z_1|z_2) \approx 1$ for a specific pair of token and close to 0 for all others. In the example in the paper, $z_1 = $ "&", and $z_2 = $ "amp". $\mathbb{P}_\mathcal{D}$ is the probability distribution induced by the pre-training data. Many-to-many bi-grams we refer to the opposite scenario where both the conditional probabilities are highly multi-modal. In the example $z_1 = $"make" and $z_2 = $"sure" we have that many other tokens can succeed $z_1 = $"make" or precede $z_2 = $"sure".

**Hit Ratios of bi-grams**    The Hit Ratio (HR@n), often referred to as hit rate, is a metric commonly used in ranking tasks. In our context, we treat each checkpoint of the language model as a "ranker" of bigrams. The Hit Ratio measures how effectively the current model checkpoint retrieves high-quality bigrams from the set of all possible bigrams. To quantify the model's progress, we define the bigrams at the final step as the "good" bigrams and measure how quickly the model approaches these high-quality bigrams. Specifically, we compute the HR@n to evaluate how often the model's output bigrams match those in the "true" top n ranked bi-grams given by the final step. Formally, the Hit Ratio@n is given by

$$\text{HR@}n = \frac{1}{n} \sum_{i=1}^{n} \mathbb{I}(\text{the i-th bigram output by the current model} \in \text{True\_Top\_n})$$

where $n$ is the number of top predictions being considered and

- $\mathbb{I}$ is the indicator function that returns 1 if the $i$-th bigram output by the model is present in the True Top $n$ bigrams, and 0 otherwise,
- True\_Top\_n represents the set of "good" bigrams, which in our case is the set of the top $n$ scoring bigrams from the final model step.

**Total Mass of Bi-grams**    We use the total mass as a metric to measure the cumulative probabilities of bi-grams from the top 1K bi-grams, weighted by an empirical unigram distribution derived from real data. Formally, it is given by: Total Mass $= \sum_{(z_1,z_2) \in \text{Top-1K}} \mathbb{P}_{e_t}(z_2|z_1) \mathbb{P}_\mathcal{D}(z_1)$ where:

- $e_t$ is the embedding-unembedding path at the $t$-th pre-training step,
- $(z_1, z_2)$ are the bigrams being considered,
- $\mathbb{P}_{e_t}(z_2|z_1)$ is the probability assigned by the model $e_t$ (the embedding-unembedding path) for the token $z_2$ given token $z_1$,
- $\mathbb{P}_\mathcal{D}(z_1)$ is the probability of $z_1$ under the empirical distribution $\mathcal{D}$, which is the unigram probability given by the Infini-gram API (**?**) on the Dolma dataset (**?**) (the dataset used to pretrain the model checkpoints).

This metric is designed to evaluate how much "correct" probability mass the model checkpoints assign to bigrams $(z_1, z_2)$, taking into account the empirical unigram probability of $z_1$. It provides insight into how well the model aligns with the empirical distribution of real-world data during the pretraining process.

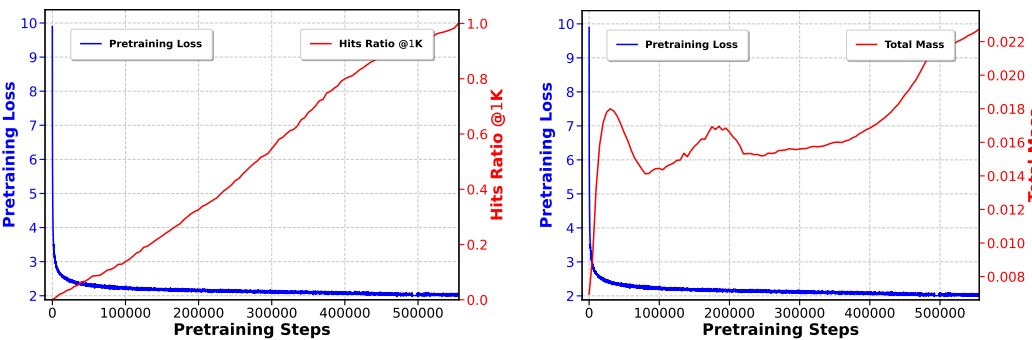

(a) Top 1K jet bi-gram hit ratios w.r.t. the final step.      (b) Top 1K jet bi-gram mass w.r.t. empirical data.

Figure 6: Analysis of *OLMo-7B*'s pretraining dynamics via measuring its jet bi-gram progression.

## E  ADDITIONAL DETAILS ON JET $n$-GRAM DIFFING

We derive the top-K bi-grams for each model from their embedding-unembedding path, which can be obtained as jet_expand($q, L, \{\eta\}, 0$). These bigrams are then saved into CSVs, allowing us to represent models via their respective bigram files. By comparing these files directly, much like comparing text files, we bypass the challenges of comparing the models in the parameter space, where measuring behavioral-level differences can be difficult. For example, we extract the bigram files for Llama-2-7B, and its coding finetuned versions. In summary, by transforming models into bigram files (Model → Bigram File), we can effectively compare their behavior via bigram file differences (Model Diff → Bigram File Diff). We will include a demonstration in supplementary material.

## F  ADDITIONAL ANALYSIS INTO THE BI-GRAMS LEARNING SPEED DURING PRETRAINING

To evaluate the learning speed of jet bi-grams during pretraining, we consider the jet bi-grams at the final training step (555K) as the ground-truth bi-grams. We then chart the hit ratios of these ground-truth bi-grams at each pretraining step, as illustrated in Figure 6a. Interestingly, even though the pretraining loss (the blue curve) shows only minor improvements after the initial 50K steps, the model's acquisition of effective bi-grams continues to progress in a steady, consistent manner. Hence bi-grams learning dynamics are active throughout the training procedure, even after the training loss stabilizes. This indicates that there is significant behavior change in the model which is not well captured by the training loss, an observation that is studied also in grokking and double-descent (Zhang et al., 2021; Power et al., 2022). In other words, jet bi-grams may offer another point of view for analyzing the learning dynamics compared to pretraining loss. In addition, fig. 6b characterizes the total pseudo-joint probability mass of top 1K bi-grams from empirical data (Liu et al., 2024). We derive a pseudo-joint jet bi-gram probability using statistical uni-grams from (Liu et al., 2024). We observe that the model gradually accumulates probability mass that aligns with the real corpus data distribution.

## G  ADDITIONAL TABLES FOR JET BI-GRAMS

See table 4 and table 5.

## H  ADDITIONAL PLOTS OF JET LENSES

See plots below, referring to the main paper for details. Note that for iterative lenses the last block coincides with the model logits for all $k$ by design. We omit the iterative lens for GPT2-large for $k = 2$ due to low cosine similarity.

Table 4: Bi-gram evolution across pretraining steps for OLMo 7B. Each column represents a distinct step, while each row corresponds to a different rank. The table entries are the bi-grams at each step for each rank. The number of tokens seen in association with the pretraining steps is also annotated. The model gradually picks up meaningful bi-grams after starting from senseless bi-grams (due to random initialization).

| Rank | 0K [#steps]
0B [#tokens] | 100K
442B | 200K
885B | 300K
1327B | 400K
1769B | 555K
2455B |
|------|------|------|------|------|------|------|
| 0 | immortal | 's | at least | &amp | &amp | &amp |
| 1 | ICUirling | at least | 's | at least | its own | its own |
| 2 | ords architect | its own | &amp | its own | their own | their own |
| 3 | yaml Adam | okerly | your own | your own | at least | his own |
| 4 | 231 next | VENT thanks | its own | their own | your own | make sure |
| 5 | clonal | iums | iums | more than | his own | your own |
| 6 | Charg@{ | you're | you're | can't | 2nd | 2nd |
| 7 | avoir careless | Everything v | 2nd | his own | more than | at least |
| 8 | HOLD worsening | erna already | you guys | 2nd | make sure | more than |
| 9 | Horse dismant | 'my | more than | make sure | can't | iums |

Table 5: The bi-grams before and after coding-finetuning. For space reason, we only show the bi-grams at every 50 ranks among the top 1000 bi-grams. We highlight the bi-grams that are relevant to coding, such as "**kwargs" a keyword in python programming. This demonstrate that our method has the capability to extract representative bi-grams that reflect fine-tuning quality.

| Rank | LLAMA2-7B | CodeLLAMA-7B | CodeLLAMA-Python-7B |
|------|-----------|--------------|---------------------|
| 0 | (_more, _than) | (_like, wise) | (_like, wise) |
| 50 | (_Now, here) | (_just, ification) | (_Like, wise) |
| 100 | (_system, atically) | (_in, _case) | (_all, udes) |
| 150 | (_all, erg) | (_get, ters) | (_no, isy) |
| 200 | (_on, ions) | (któber, s) | (output, ted) |
| 300 | (_other, world) | (_all, ud) | (Object, ive) |
| 350 | (_Just, ified) | (gebiet, s) | (_as, cii) |
| 400 | (_trust, ees) | (_Protest, s) | (_can, nab) |
| 450 | (_at, he) | (_deploy, ment) | (_transport, ation) |
| 500 | (_book, mark) | (Class, room) | (Tag, ging) |
| 550 | (_from, ) | (_access, ory) | (_personal, ized) |
| 600 | (_WHEN, ever) | (_In, variant) | (_excess, ive) |
| 650 | (_where, about) | (_I, _am) | (_Add, itional) |
| 700 | (ag, ged) | (add, itionally) | (_**, kwargs) |
| 750 | (_he, he) | (_invalid, ate) | (name, plates) |
| 800 | (_all, anto) | (div, ision) | (_select, ive) |
| 850 | (_Tom, orrow) | (_process, ors) | (_Assert, ions) |
| 900 | (_for, ays) | (_Program, me) | (blog, ger) |
| 950 | (_Bach, elor) | (_set, up) | (_can, cellation) |

Figure 7: Iterative jet lens ($k = 0$), equivalent to logit lens(nostalgebraist, 2021b), applied over GPT-Neo-2.7B with the input sentence "new simple neural architecture, the Transformer".

| | new | _simple | _neural | _architecture | , | _the | _Trans | former |
|---|---|---|---|---|---|---|---|---|
| Block 1 | , | ton | _network | _for | _which | _first | former | , |
| Block 2 | Supporters | ton | _network | _for | _which | _first | former | , |
| Block 3 | Supporters | ton | _network | _for | _which | _first | former | , |
| Block 4 | Supporters | ton | _network | _for | _which | _first | former | , |
| Block 5 | Supporters | ton | _network | _for | _which | _first | former | , |
| Block 6 | Supporters | ton | _network | _for | _which | _first | former | , |
| Block 7 | Supporters | ton | _network | _for | _which | _first | former | , |
| Block 8 | Supporters | ton | _network | _for | _which | _first | former | , |
| Block 9 | Supporters | ton | _network | _for | _which | _first | former | , |
| Block 10 | Supporters | ton | _network | _for | _which | _first | former | , |
| Block 11 | Supporters | ton | _network | _for | _which | _first | former | , |
| Block 12 | Supporters | ton | _network | _for | _which | _first | former | , |
| Block 13 | Supporters | ton | _network | _for | _which | _first | former | , |
| Block 14 | Supporters | ton | _network | _for | _which | _first | former | , |
| Block 15 | Supporters | ton | _network | _for | _which | _first | former | , |
| Block 16 | Supporters | ton | _network | _for | _which | _first | former | , |
| Block 17 | Supporters | ton | _network | _for | _which | _first | former | , |
| Block 18 | Supporters | ton | _network | _for | _which | _first | former | , |
| Block 19 | Supporters | ton | _network | _for | _which | _first | former | , |
| Block 20 | Supporters | ton | _network | _for | _which | _first | former | , |
| Block 21 | Supporters | ton | _network | _for | _which | _first | former | , |
| Block 22 | Supporters | ton | _network | _for | _which | _first | former | , |
| Block 23 | Supporters | ton | _network | _for | _which | _first | former | , |
| Block 24 | Supporters | ton | _network | _for | _which | _so | former | , |
| Block 25 | Supporters | ton | _network | _for | _which | _first | former | , |
| Block 26 | Supporters | ton | _network | _for | _which | _first | former | , |
| Block 27 | Supporters | ton | _network | _for | _which | _first | former | , |
| Block 28 | Supporters | ton | _network | _for | _which | _first | former | , |
| Block 29 | foreseen | ton | _network | _for | _which | _first | former | , |
| Block 30 | foreseen | ton | _network | _for | _which | _first | former | , |
| Block 31 | Supporters | _ | _network | _for | _which | _first | former | , |
| Block 32 | _ | _ | _network | _for | _which | _neural | former | , |
| | | | | | | | | |
| Logits | _ | _ | _network | _for | _which | _neural | former | , |

Figure 8: Iterative jet lens ($k = 1$), applied over GPT-Neo-2.7B with the input sentence "new simple neural architecture, the Transformer"

| | new | _simple | _neural | _architecture | , | _the | _Trans | former |
|---|---|---|---|---|---|---|---|---|
| Block 1 | _the | _ | _nets | !: | _âĢ¦" | _âĢ¦" | former | !: |
| Block 2 | _the | _ | _network | _outper | _âĢ¦" | _âĢ¦" | former | _[ |
| Block 3 | _the | _ | _network | _for | _trained | _Conv | former | _[ |
| Block 4 | _the | _ | _network | _for | _the | _Conv | former | , |
| Block 5 | _the | _ | _network | _for | _the | _neural | former | , |
| Block 6 | _the | _ | _network | _for | _the | _neural | former | , |
| Block 7 | _the | _ | _network | _for | _the | architecture | former | , |
| Block 8 | _the | _ | _network | _for | _the | architecture | former | , |
| Block 9 | _the | _ | _network | _for | _the | architecture | former | , |
| Block 10 | _the | _ | _network | _for | _the | architecture | former | , |
| Block 11 | _the | _ | _network | _for | _the | architecture | former | , |
| Block 12 | _the | _ | _network | _for | _the | architecture | former | , |
| Block 13 | _the | _ | _network | _for | _the | architecture | former | , |
| Block 14 | _the | _ | _network | _for | _the | _neural | former | , |
| Block 15 | _the | _ | _network | _for | _the | _neural | former | , |
| Block 16 | _the | _ | _network | _for | _the | _neural | former | , |
| Block 17 | _the | _ | _network | _for | _the | _neural | former | , |
| Block 18 | _the | _ | _network | _for | _the | _neural | former | , |
| Block 19 | _the | _ | _network | _for | _the | _neural | former | , |
| Block 20 | _the | _ | _network | _for | _the | _neural | former | , |
| Block 21 | _the | _ | _network | _for | _the | _neural | former | , |
| Block 22 | _the | _ | _network | _for | _the | _neural | former | , |
| Block 23 | _the | _ | _network | _for | _the | _neural | former | , |
| Block 24 | _the | _ | _network | _for | _the | _neural | former | , |
| Block 25 | _the | _ | _network | _for | _the | _neural | former | , |
| Block 26 | _the | _ | _network | _for | _the | _neural | former | , |
| Block 27 | _the | _ | _network | _for | _the | _neural | former | , |
| Block 28 | _the | _ | _network | _for | _the | _neural | former | , |
| Block 29 | _the | _ | _network | _for | _the | _neural | former | , |
| Block 30 | _the | _ | _network | _for | _and | _neural | former | , |
| Block 31 | , | _ | _network | _for | _and | _neural | former | , |
| Block 32 | _ | _ | _network | _for | _which | _neural | former | , |
| | | | | | | | | |
| Logits | _ | _ | _network | _for | _which | _neural | former | , |

Figure 9: Iterative jet lens ($k = 2$), applied over GPT-Neo-2.7B with the input sentence "new simple neural architecture, the Transformer"

| | new | _simple | _neural | _architecture | , | _the | _Trans | former |
|---|---|---|---|---|---|---|---|---|
| Block 1 | bie | _simple | _neural | _architecture | _and | _the | fig | former |
| Block 2 | bie | _simple | _neural | _architecture | _and | _main | ient | former |
| Block 3 | bie | _simple | _neural | _architecture | _and | _new | ient | former |
| Block 4 | bie | _way | _neural | _architecture | _and | _first | ient | _titan |
| Block 5 | bie | _way | _networks | _architecture | _and | _next | ient | _Prime |
| Block 6 | bie | _enough | _networks | _architecture | _and | _next | ient | _Matrix |
| Block 7 | _href | _enough | _networks | _architecture | _and | _first | ient | _Prime |
| Block 8 | _iTunes | _enough | _neural | _architecture | _which | _first | ient | _Revolution |
| Block 9 | , | _enough | _neural | _architecture | _which | _first | ient | _Prime |
| Block 10 | , | _enough | _network | _architecture | _which | _first | ient | _Revolution |
| Block 11 | , | _enough | _network | _model | _which | _only | ient | _Pro |
| Block 12 | , | _enough | _network | _architecture | _which | _only | ient | _Pro |
| Block 13 | , | _enough | _network | _model | _which | _first | ient | _Pro |
| Block 14 | , | _enough | _network | _model | _which | _first | ient | _Pro |
| Block 15 | , | _enough | _network | _model | _which | _only | ient | _Pro |
| Block 16 | , | - | _network | _model | _which | _only | ient | _Revolution |
| Block 17 | , | - | _system | _model | _which | _only | ient | _Prime |
| Block 18 | , | - | _system | _model | _which | _only | ient | _Prime |
| Block 19 | , | - | _system | _model | _which | _only | ient | _Prime |
| Block 20 | , | - | _system | _model | _which | _only | ient | _Prime |
| Block 21 | , | - | _system | _model | _which | _only | ient | _Prime |
| Block 22 | , | - | _network | _model | _which | _only | ient | _Prime |
| Block 23 | , | ton | _network | _model | _which | _only | ient | _Prime |
| Block 24 | , | ton | _network | _model | _which | _only | ient | _Prime |
| Block 25 | , | ton | _network | _model | _which | _first | ient | _Prime |
| Block 26 | , | ton | _network | _model | _which | _only | ient | _Prime |
| Block 27 | , | ton | _network | _for | _which | _first | ient | _Prime |
| Block 28 | , | - | _network | " | _which | _only | ient | _Prime |
| Block 29 | , | - | _network | " | _which | _neural | ient | _Prime |
| Block 30 | , | " | _network | " | _which | _neural | ient | , |
| Block 31 | , | " | _network | " | _which | _neural | ient | , |
| Block 32 | , | " | _network | " | _which | _neural | ient | , |
| Block 33 | , | " | _network | _for | _which | _neural | ient | , |
| Block 34 | , | " | _network | ' | _which | _neural | ient | , |
| Block 35 | , | " | _network | ' | _which | _neural | c | , |
| Block 36 | _ | " | _network | ' | _which | _neural | c | , |
| | | | | | | | | |
| Logits | _ | " | _network | ' | _which | _neural | c | , |

Figure 10: Iterative jet lens ($k = 0$), equivalent to logit lens(nostalgebraist, 2021b), applied over GPT-2-large with the input sentence "new simple neural architecture, the Transformer".

| | new | _simple | _neural | _architecture | , | _the | _Trans | former |
|---|---|---|---|---|---|---|---|---|
| Block 1 | bie | " | _network | " | _which | _neural | c | _is |
| Block 2 | bie | " | _network | ' | _which | _neural | c | _is |
| Block 3 | bie | " | _network | ' | _which | _neural | c | _is |
| Block 4 | _ | " | _network | ' | _which | _neural | c | _is |
| Block 5 | _ | " | _network | ' | _which | _neural | c | _is |
| Block 6 | _ | " | _network | ' | _which | _neural | c | _is |
| Block 7 | _ | " | _network | ' | _which | _neural | c | _is |
| Block 8 | _ | " | _network | ' | _which | _neural | c | _is |
| Block 9 | _ | " | _network | ' | _which | _neural | c | _is |
| Block 10 | , | " | _network | ' | _which | _neural | c | _is |
| Block 11 | , | " | _network | ' | _which | _neural | c | _is |
| Block 12 | , | " | _network | ' | _which | _neural | c | , |
| Block 13 | , | " | _network | ' | _where | _neural | c | , |
| Block 14 | , | " | _network | ' | _and | _neural | c | , |
| Block 15 | , | " | _network | ' | _and | _neural | c | , |
| Block 16 | , | " | _network | ' | _and | _neural | c | , |
| Block 17 | , | " | _network | ' | _and | _neural | c | , |
| Block 18 | , | " | _network | ' | _and | _neural | c | , |
| Block 19 | , | " | _network | ' | _and | _neural | c | , |
| Block 20 | , | " | _network | ' | _and | _neural | c | , |
| Block 21 | , | " | _network | ' | _and | _neural | c | , |
| Block 22 | , | " | _network | ' | _and | _neural | c | , |
| Block 23 | , | " | _network | ' | _the | _neural | c | , |
| Block 24 | , | " | _network | ' | _and | _neural | c | , |
| Block 25 | , | " | _network | ' | _and | _neural | c | , |
| Block 26 | , | " | _network | ' | _and | _neural | c | , |
| Block 27 | , | " | _network | ' | _and | _neural | c | , |
| Block 28 | , | " | _network | ' | _and | _neural | c | , |
| Block 29 | , | " | _network | ' | _and | _human | c | , |
| Block 30 | , | " | _network | ' | _and | _same | c | , |
| Block 31 | , | " | _network | ' | _and | _same | c | , |
| Block 32 | , | " | _network | ' | _and | _same | c | , |
| Block 33 | , | " | _network | ' | _and | _neural | c | , |
| Block 34 | , | - | _network | ' | _which | _neural | c | , |
| Block 35 | - | " | _network | ' | _which | _neural | c | , |
| Block 36 | _ | " | _network | ' | _which | _neural | c | , |
| | | | | | | | | |
| Logits | _ | " | _network | ' | _which | _neural | c | , |

Figure 11: Iterative jet lens ($k = 1$), applied over GPT-2-large with the input sentence "new simple neural architecture, the Transformer"

|  | new | _simple | _neural | _architecture | , | _the | _Trans | former |
|---|---|---|---|---|---|---|---|---|
| Block 1 (4.40%) | , (6.62%) | _simple (3.91%) | _neural (4.42%) | _architecture (3.97%) | _which (4.07%) | _same (4.37%) | cend (3.93%) | former (3.91%) |
| Block 2 (4.15%) | , (6.59%) | _retro (3.85%) | _prog (4.32%) | _error (3.74%) | _including (3.93%) | _resulting (4.14%) | ference (3.69%) | _Robo (2.99%) |
| Block 3 (4.23%) | , (6.59%) | ove (4.13%) | _Matter (4.12%) | killer (3.51%) | _which (4.00%) | _AVG (4.01%) | em (3.56%) | Mars (3.91%) |
| Block 4 (4.11%) | _the (6.59%) | _reg (3.51%) | lect (4.37%) | OX (3.68%) | _found (4.05%) | netflix (4.09%) | Charge (2.95%) | Â® (3.69%) |
| Block 5 (6.11%) | , (6.59%) | ware (3.54%) | _product (3.68%) | _towards (3.70%) | _evolution (3.88%) | _ones (3.74%) | it (20.20%) | _Mant (3.57%) |
| Block 6 (3.91%) | , (6.58%) | ies (3.59%) | _networks (4.11%) | _developed (3.45%) | _developed (3.55%) | _Mehran (3.45%) | ition (3.54%) | bur (3.01%) |
| Block 7 (4.00%) | , (6.56%) | face (3.75%) | _studies (3.88%) | _based (3.52%) | _hackers (3.76%) | _Turing (3.73%) | _Series (2.97%) | _Suite (3.83%) |
| Block 8 (4.06%) | , (6.42%) | key (3.83%) | _model (4.18%) | _based (3.53%) | _requiring (3.49%) | _algorithm (4.14%) | ient (3.62%) | _II (3.25%) |
| Block 9 (4.09%) | , (7.45%) | _clutter (4.08%) | _model (3.69%) | _test (3.40%) | _which (3.11%) | _neural (3.55%) | verse (3.82%) | _Cube (3.66%) |
| Block 10 (10.50%) | . (16.50%) | lists (9.61%) | g (4.99%) | _of (16.60%) | _which (11.47%) | _neural (5.79%) | _neural (3.50%) | _is (15.56%) |
| Block 11 (25.30%) | , (16.96%) | " (27.59%) | _networks (28.89%) | " (24.52%) | _the (26.92%) | _new (29.14%) | m (22.95%) | _neural (25.40%) |
| Block 12 (25.13%) | , (6.56%) | . (28.62%) | net (29.35%) | , (26.40%) | the (27.77%) | the (29.85%) | c (25.27%) | . (27.23%) |
| Logits | , | - | _network | _that | _which | _neural | ient | _is |
| Expan. (1.000) | , | - | _network | _of | _which | " | - | _is |

Figure 12: Joint jet lens with learnable weightings ($k = 0$), applied over GPT2 with the input sentence "new simple neural architecture, the Transformer"

|  | new | _simple | _neural | _architecture | , | _the | _Trans | former |
|---|---|---|---|---|---|---|---|---|
| Block 1 (15.30%) | . (7.49%) | " (16.78%) | _networks (16.96%) | ", (18.37%) | _neural (14.61%) | _neural (14.05%) | verse (16.45%) | _Neural (17.73%) |
| Block 2 (4.57%) | , (13.81%) | json (3.21%) | _networks (3.29%) | _model (3.46%) | _which (3.11%) | _neural (3.02%) | cend (3.23%) | _Neural (3.45%) |
| Block 3 (4.49%) | , (14.25%) | tons (3.25%) | _networks (2.82%) | _architecture (3.32%) | _neural (3.10%) | _neural (3.00%) | porter (3.03%) | _Neural (3.17%) |
| Block 4 (4.10%) | . (11.55%) | tons (3.28%) | _networks (3.27%) | _leveraging (3.19%) | _synt (3.04%) | _neural (2.98%) | verse (2.90%) | _Neural (2.57%) |
| Block 5 (4.02%) | . (9.58%) | tons (3.05%) | _networks (3.25%) | _algorithm (3.45%) | _which (3.14%) | _neural (2.99%) | mitter (3.24%) | _Neural (3.47%) |
| Block 6 (3.02%) | . (2.75%) | _linkage (2.65%) | _net (3.04%) | _algorithms (3.26%) | _detecting (2.94%) | _neural (2.80%) | cend (3.30%) | _Neural (3.45%) |
| Block 7 (2.91%) | . (2.98%) | _teleportation (2.78%) | _nets (3.19%) | _approach (3.24%) | _specifically (2.49%) | _cortex (2.53%) | genic (3.07%) | _Cortex (2.95%) |
| Block 8 (4.60%) | bid (3.10%) | nex (7.64%) | _network (2.63%) | _platform (2.62%) | _neural (4.81%) | _participant (9.06%) | cription (3.50%) | _Neural (3.45%) |
| Block 9 (7.44%) | iaries (3.10%) | url (5.60%) | _networks (7.77%) | _intelligence (4.86%) | _Torch (14.64%) | _welcoming (13.48%) | Secure (7.21%) | _conv (2.83%) |
| Block 10 (15.04%) | akings (13.99%) | widget (14.80%) | _network (16.20%) | _None (13.05%) | _Bund (15.37%) | _safest (14.72%) | cend (16.11%) | _disabling (16.06%) |
| Block 11 (16.50%) | ity (3.19%) | ton (18.47%) | _network (18.79%) | _architecture (20.49%) | _which (16.34%) | _neural (15.62%) | istor (18.84%) | â††¢ (20.28%) |
| Block 12 (18.00%) | , (14.21%) | - (18.49%) | network (18.78%) | that (20.68%) | which (16.41%) | neural (15.70%) | ient (19.11%) | is (20.60%) |
| Logits | , | - | _network | _that | _which | _neural | ient | _is |
| Expan. (1.000) | akings | json | _networks | _framework | _neural | _neural | cend | _Neural |

Figure 13: Joint jet lens with learnable weightings ($k = 1$), applied over GPT2 with the input sentence "new simple neural architecture, the Transformer"

|  | new | _simple | _neural | _architecture | , | _the | _Trans | former |
|---|---|---|---|---|---|---|---|---|
| Block 1 (3.58%) | Supporters (1.55%) | Supporters (3.24%) | Supporters (3.46%) | Supporters (5.37%) | Supporters (5.08%) | Supporters (3.52%) | Supporters (3.88%) | Supporters (2.56%) |
| Block 2 (2.13%) | foreseen (1.61%) | foreseen (2.97%) | foreseen (1.15%) | Introduced (3.96%) | foreseen (1.09%) | foreseen (1.54%) | Supporters (3.67%) | Supporters (1.03%) |
| Block 3 (2.07%) | Amid (1.65%) | Supporters (2.01%) | Across (1.32%) | gewater (1.14%) | Supporters (3.66%) | Supporters (2.93%) | Supporters (2.58%) | leground (1.28%) |
| Block 4 (1.57%) | _impover (1.97%) | _unpop (2.18%) | _unpop (1.46%) | _impover (1.33%) | _impover (1.39%) | _impover (1.71%) | _uphe (1.27%) | _impover (1.27%) |
| Block 5 (1.47%) | Attempts (1.76%) | _municip (2.15%) | _airst (1.45%) | _linem (1.29%) | amiliar (1.32%) | pelling (1.38%) | rieving (1.26%) | _linem (1.13%) |
| Block 6 (1.45%) | Residents (1.76%) | _athlet (2.17%) | rha (1.44%) | _twent (1.34%) | _way (1.05%) | ters (1.40%) | rha (1.23%) | _Xuan (1.25%) |
| Block 7 (3.57%) | Ironically (1.63%) | celona (2.74%) | wrap (3.78%) | _look (5.71%) | _airstrike (1.22%) | _equivalent (2.63%) | _different (6.30%) | _hollow (4.58%) |
| Block 8 (4.63%) | Supporters (1.61%) | imura (3.03%) | vantage (3.03%) | anoia (5.48%) | foreseen (6.13%) | ileen (4.55%) | Enlarge (5.70%) | assador (6.59%) |
| Block 9 (3.14%) | Ironically (1.65%) | erguson (2.00%) | certain (2.53%) | OUR (1.28%) | _local (3.54%) | erguson (1.80%) | enter (5.43%) | bec (6.89%) |
| Block 10 (1.73%) | foreseen (1.65%) | foreseen (2.01%) | Engineers (1.20%) | Engineers (1.20%) | asury (1.19%) | thinkable (1.40%) | Attempts (2.53%) | uddenly (0.96%) |
| Block 11 (1.71%) | likely (1.57%) | extremely (1.88%) | aples (1.18%) | _screenplay (1.29%) | earances (1.30%) | earances (4.13%) | oother (1.20%) | _resurg (1.12%) |
| Block 12 (4.53%) | Ironically (1.73%) | Phones (3.91%) | ADVERTISEMENT (4.39%) | ADVERTISEMENT (6.03%) | isively (4.15%) | _Blvd (4.46%) | ADVERTISEMENT (6.08%) | ADVERTISEMENT (4.99%) |
| Block 13 (2.80%) | _a (1.68%) | aji (2.83%) | imbabwe (1.33%) | rone (1.28%) | OTOS (5.38%) | ppard (3.08%) | ppard (1.07%) | aji (5.76%) |
| Block 14 (2.91%) | foreseen (1.66%) | ADVERTISEMENT (1.83%) | Marginal (3.82%) | chell (1.32%) | _Appalach (1.33%) | _Caucasus (4.66%) | _still (5.47%) | , (3.23%) |
| Block 15 (1.47%) | ormons (1.78%) | _confir (1.89%) | uring (1.34%) | ured (1.25%) | AoE (1.38%) | _Caucas (1.68%) | _lineman (1.25%) | _topple (1.22%) |
| Block 16 (3.98%) | Against (1.82%) | folios (1.93%) | @ (6.49%) | thinkable (3.49%) | _tsun (1.26%) | _D (4.65%) | I (5.84%) | arsh (6.38%) |
| Block 17 (2.89%) | urses (1.38%) | untied (4.46%) | ortunate (3.72%) | ithub (1.21%) | _our (4.69%) | ortment (1.51%) | erenn (4.91%) | ombies (1.21%) |
| Block 18 (5.12%) | foreseen (1.63%) | Supporters (4.53%) | Nonetheless (6.62%) | Ironically (5.07%) | Thankfully (5.66%) | Shortly (4.52%) | af (5.80%) | _is (7.12%) |
| Block 19 (2.96%) | pherd (1.47%) | _enough (4.91%) | ag (3.58%) | _for (5.69%) | incerity (1.08%) | incerity (2.75%) | extreme (3.01%) | phabet (1.21%) |
| Block 20 (5.68%) | Ć (2.06%) | Ć (5.07%) | _just (7.05%) | Ć (6.91%) | Attempts (6.51%) | paralleled (4.49%) | - (6.53%) | , (6.87%) |
| Block 21 (1.06%) | ription (1.60%) | ription (2.15%) | _Playoffs (1.48%) | isdom (1.06%) | _frontrunner (1.36%) | _frontrunner (1.69%) | _TBD (1.24%) | pered (1.06%) |
| Block 22 (4.55%) | _in (3.36%) | _first (5.29%) | _two (7.06%) | _one (6.98%) | _which (6.97%) | _one (4.56%) | _isEnabled (1.03%) | elligence (1.15%) |
| Block 23 (5.21%) | , (4.80%) | 1) (5.23%) | _' (7.13%) | ) (6.26%) | _while (6.31%) | _point (4.57%) | albeit (1.15%) | B (6.21%) |
| Block 24 (6.13%) | _a (5.62%) | _m (5.26%) | _first (7.18%) | _for (7.33%) | _the (7.33%) | _so (4.70%) | _trans (5.70%) | rieving (5.90%) |
| Block 25 (1.55%) | foreseen (1.67%) | acly (2.14%) | _enthus (1.49%) | _anecd (1.35%) | _trainers (1.43%) | _subreddits (1.74%) | ithub (1.28%) | _Trainer (1.27%) |
| Block 26 (2.61%) | - (6.25%) | _simple (2.08%) | _simple (5.95%) | ername (1.30%) | haar (1.34%) | _satell (1.74%) | igsaw (1.02%) | _headphone (1.17%) |
| Block 27 (2.65%) | âĞ (7.40%) | âĞ (5.48%) | _DSM (1.35%) | held (1.30%) | dayName (1.38%) | _artif (1.75%) | --+ (1.27%) | _nostalg (1.30%) |
| Block 28 (2.39%) | _fps (8.56%) | >>\ (2.30%) | _Oo (1.42%) | _tacos (1.30%) | _rnsec (1.41%) | _unbeliev (1.75%) | _hrs (1.12%) | _reminis (1.28%) |
| Block 29 (1.97%) | âĞ¦ (5.17%) | _convol (2.18%) | ricanes (1.47%) | _Gujar (1.25%) | acerb (1.38%) | cffff (1.74%) | _negoti (1.28%) | _automakers (1.27%) |
| Block 30 (1.84%) | âĞ¦ (4.01%) | _anecd (2.24%) | _unve (1.49%) | _overwhel (1.37%) | !?* (1.43%) | 20439 (1.78%) | _negoti (1.29%) | _calculates (1.12%) |
| Block 31 (4.61%) | !!* (8.40%) | âĞ¦ (2.57%) | _greets (1.35%) | _entert (1.80%) | \\\ (4.44%) | \\\ (6.14%) | *! (5.27%) | '/ (6.88%) |
| Block 32 (5.64%) | âĞ¦," (9.55%) | !?* (4.42%) | âĞ¦," (2.29%) | âĞ¦," (5.37%) | âĞ¦" (6.35%) | _\' (9.03%) | ®¶æ¥µ (3.34%) | âĞ¦," (4.75%) |
| Logits | _ |  | _network | _for | _which | _neural | former | , |
| Expan. (0.977) | _the | _and | - | _for | _the | _first | - | , |

Figure 14: Joint jet lens with learnable weightings ($k = 0$), applied over GPT-Neo-2.7B with the input sentence "new simple neural architecture, the Transformer"

|  | new | _simple | _neural | _architecture | , | _the | _Trans | former |
|---|---|---|---|---|---|---|---|---|
| Block 1 (7.36%) | , (3.40%) | ton (8.06%) | _network (8.57%) | _for (8.22%) | _which (7.51%) | _first (7.30%) | former (7.43%) | , (8.36%) |
| Block 2 (4.83%) | - (2.39%) | _ (5.23%) | _network (6.91%) | _for (4.98%) | _which (4.60%) | _neural (4.77%) | former (5.09%) | , (4.68%) |
| Block 3 (1.31%) | _File (1.62%) | _ (1.29%) | _network (1.31%) | _for (1.28%) | _which (1.25%) | _CNN (1.22%) | former (1.20%) | , (1.32%) |
| Block 4 (7.81%) | _impover (5.74%) | _unpop (8.48%) | _impover (8.76%) | _impover (8.45%) | _impover (7.67%) | _Neural (7.51%) | former (7.39%) | _Networks (8.45%) |
| Block 5 (1.79%) | User (5.29%) | _ (1.31%) | _network (1.30%) | _for (1.29%) | _which (1.29%) | _neural (1.26%) | former (1.25%) | , (1.31%) |
| Block 6 (1.79%) | Instance (5.33%) | _ (1.33%) | _network (1.31%) | _for (1.29%) | _which (1.26%) | _neural (1.23%) | former (1.23%) | , (1.32%) |
| Block 7 (1.59%) | File (3.56%) | _ (1.37%) | _network (1.36%) | _for (1.33%) | _which (1.28%) | _neural (1.24%) | former (1.25%) | , (1.32%) |
| Block 8 (1.70%) | Supporters (5.02%) | _ (1.29%) | _network (1.28%) | _for (1.25%) | _which (1.24%) | _Neural (1.17%) | former (1.12%) | , (1.21%) |
| Block 9 (1.77%) | Enlarge (5.04%) | _ (1.37%) | _network (1.37%) | _for (1.32%) | _which (1.26%) | _neural (1.23%) | former (1.25%) | , (1.31%) |
| Block 10 (4.41%) | foreseen (5.36%) | _ (5.77%) | _network (6.19%) | _for (5.99%) | _which (1.15%) | _neural (0.93%) | former (2.45%) | , (7.42%) |
| Block 11 (1.31%) | , (1.90%) | _ (1.30%) | _network (1.29%) | _for (1.20%) | _which (1.18%) | _neural (1.19%) | former (1.19%) | , (1.24%) |
| Block 12 (1.21%) | , (1.74%) | _ (1.11%) | _network (1.17%) | _for (1.10%) | _which (1.16%) | _neural (1.15%) | former (1.07%) | , (1.21%) |
| Block 13 (1.37%) | _ (1.94%) | _ (1.36%) | _network (1.35%) | _for (1.32%) | _which (1.23%) | _neural (1.21%) | former (1.23%) | , (1.32%) |
| Block 14 (1.22%) | , (1.82%) | _ (1.18%) | _network (1.22%) | _for (1.12%) | _which (1.15%) | _neural (1.09%) | former (1.04%) | , (1.12%) |
| Block 15 (1.34%) | _ (1.90%) | _ (1.33%) | _network (1.31%) | _for (1.29%) | _which (1.21%) | _neural (1.20%) | former (1.20%) | , (1.28%) |
| Block 16 (1.31%) | ( (1.91%) | _ (1.28%) | _network (1.28%) | _for (1.24%) | _which (1.18%) | _neural (1.19%) | former (1.18%) | _model (1.23%) |
| Block 17 (1.31%) | _ (1.90%) | _ (1.29%) | _network (1.28%) | _for (1.26%) | _which (1.14%) | _neural (1.12%) | former (1.16%) | , (1.29%) |
| Block 18 (4.55%) | , (1.65%) | _ (5.16%) | _network (3.55%) | _for (5.49%) | _which (6.28%) | _neural (6.05%) | former (5.05%) | , (3.17%) |
| Block 19 (1.24%) | , (1.84%) | _ (1.23%) | _network (1.17%) | _for (1.18%) | _which (1.23%) | _neural (0.97%) | former (1.10%) | _model (1.18%) |
| Block 20 (3.30%) | Ċ (1.84%) | _ (2.30%) | _network (1.16%) | _for (4.21%) | _which (6.29%) | _neural (5.89%) | former (2.70%) | _architecture (2.00%) |
| Block 21 (1.87%) | _ (1.80%) | _ (1.21%) | _network (1.12%) | _for (1.15%) | _which (3.82%) | _neural (3.71%) | former (1.10%) | , (1.02%) |
| Block 22 (4.81%) | - (1.91%) | _infographic (8.14%) | _network (3.50%) | _outper (5.92%) | _which (6.89%) | _neural (6.76%) | former (1.57%) | _[ (3.83%) |
| Block 23 (2.01%) | , (1.91%) | _ (1.14%) | _network (1.40%) | _learns (1.38%) | _which (3.94%) | _Conv (3.99%) | former (1.14%) | _model (1.18%) |
| Block 24 (6.02%) | , (1.94%) | _infographic (8.04%) | _network (7.20%) | _unve (8.00%) | _unve (7.47%) | _Neural (7.02%) | former (3.53%) | _model (4.98%) |
| Block 25 (1.19%) | _ (1.87%) | _ (1.19%) | _network (1.09%) | _for (1.22%) | _which (0.96%) | _âG (1.07%) | former (1.06%) | , (1.04%) |
| Block 26 (1.55%) | _ (1.89%) | _ (1.18%) | _network (2.18%) | _called (1.22%) | _which (1.25%) | _Conv (1.09%) | former (2.57%) | , (1.06%) |
| Block 27 (2.23%) | _ (1.93%) | ton (3.53%) | _network (1.09%) | _for (1.21%) | _which (0.99%) | _model (1.13%) | former (6.67%) | , (1.25%) |
| Block 28 (2.76%) | _ (1.73%) | json (1.02%) | _network (3.49%) | _for (1.84%) | _which (0.95%) | _Neural (3.31%) | former (6.31%) | , (3.42%) |
| Block 29 (3.22%) | âG¦" (6.01%) | _ (1.32%) | _network (1.00%) | _for (1.01%) | _and (1.74%) | _neural (1.90%) | former (7.25%) | , (5.54%) |
| Block 30 (6.24%) | âG¦" (6.04%) | _ (3.56%) | _network (7.34%) | _for (5.45%) | _which (6.05%) | _neural (6.14%) | former (7.30%) | Äí (8.04%) |
| Block 31 (7.76%) | !!" (5.96%) | _ (8.27%) | _network (8.68%) | _for (8.36%) | _the (7.67%) | _Conv (7.46%) | former (7.35%) | , (8.37%) |
| Block 32 (7.84%) | âG¦." (5.81%) | !?* (8.35%) | _network (8.78%) | , (8.43%) | _and (7.70%) | _neural (7.51%) | former (7.57%) | _model (8.53%) |
| Logits | _ | _ | _network | _for | _which | _neural | former | , |
| Expan. (0.993) | _ | _ | _network | _for | _which | _neural | former | , |

Figure 15: Joint jet lens with learnable weightings ($k = 1$), applied over GPT-Neo-2.7B with the input sentence "new simple neural architecture, the Transformer"

|  | new | _simple | _neural | _architecture | , | _the | _Trans | former |
|---|---|---|---|---|---|---|---|---|
| Block 1 (3.19%) | bie (4.48%) | _simple (4.99%) | _neural (0.98%) | _architecture (1.08%) | _and (5.08%) | _the (5.85%) | fig (2.07%) | former (1.01%) |
| Block 2 (1.81%) | _arrivals (2.43%) | tons (1.22%) | _rack (3.83%) | _model (1.07%) | _the (1.01%) | _main (1.01%) | ient (3.10%) | _generation (0.85%) |
| Block 3 (2.49%) | _entry (5.53%) | _fitting (5.41%) | _clusters (3.05%) | _det (1.14%) | _thanks (0.99%) | _second (1.00%) | cription (0.97%) | _barrier (1.86%) |
| Block 4 (3.02%) | bies (3.47%) | _private (5.64%) | _env (5.41%) | _clusters (1.15%) | _aspirin (1.09%) | _hypothesis (1.08%) | cript (5.55%) | _Mund (0.75%) |
| Block 5 (1.75%) | _mansion (3.47%) | _Transcript (1.03%) | ous (2.48%) | _suit (1.15%) | chuk (1.11%) | _Oracle (1.17%) | _Card (2.55%) | cknow (1.00%) |
| Block 6 (1.84%) | _Parables (2.46%) | _Bald (1.45%) | izer (0.99%) | sche (1.21%) | %); (1.11%) | ija (1.18%) | ione (5.34%) | atti (1.01%) |
| Block 7 (2.51%) | DERR (2.47%) | _sp (1.62%) | _wired (3.21%) | inea (1.19%) | )* (1.02%) | _gloss (1.17%) | aways (4.96%) | _system (4.48%) |
| Block 8 (1.80%) | , (2.32%) | _Tall (1.04%) | _experiments (0.89%) | MIT (1.21%) | mac (1.06%) | fts (1.16%) | rock (5.75%) | con (0.97%) |
| Block 9 (1.79%) | , (2.19%) | onel (1.11%) | _layer (5.70%) | _hum (1.10%) | arily (1.06%) | _Hots (1.20%) | iter (0.98%) | _boxes (0.96%) |
| Block 10 (2.17%) | , (2.18%) | tested (1.09%) | / (6.21%) | _deployed (1.18%) | _disrupt (3.01%) | ew (1.11%) | _INS (0.76%) | _Drive (1.80%) |
| Block 11 (1.20%) | , (2.18%) | azon (1.10%) | âh²âH_ (1.00%) | ea (1.20%) | Ro (1.10%) | _Dive (1.10%) | _Revised (0.95%) | _Prol (1.00%) |
| Block 12 (1.17%) | , (2.20%) | _Think (1.05%) | _Dish (0.86%) | _Layer (1.11%) | _Sing (0.99%) | uts (1.16%) | _button (0.94%) | _proble (1.02%) |
| Block 13 (1.88%) | _and (2.22%) | _ab (2.77%) | ourt (4.71%) | _Malf (1.20%) | _REPL (0.99%) | _naked (1.17%) | oran (0.98%) | _cred (1.01%) |
| Block 14 (1.60%) | _and (2.22%) | alg (1.06%) | _underestimated (0.97%) | _percentile (1.19%) | _which (2.35%) | _nonetheless (1.15%) | igo (3.05%) | _Hut (0.81%) |
| Block 15 (2.19%) | _and (2.24%) | - (4.45%) | _Subst (1.01%) | chan (1.16%) | ATURES (1.09%) | _hitch (1.19%) | _Mini (0.99%) | _Bre (5.41%) |
| Block 16 (2.24%) | _and (2.26%) | _image (5.83%) | _cell (4.89%) | _packs (1.05%) | _marked (0.91%) | _Finn (1.09%) | omes (0.89%) | _Cipher (0.99%) |
| Block 17 (1.72%) | _and (2.27%) | Äɬ (1.11%) | _formulation (0.96%) | isen (1.22%) | _modular (1.08%) | _Space (0.99%) | _Neural (0.85%) | _Trainer (5.29%) |
| Block 18 (1.54%) | _and (2.21%) | _bond (1.06%) | _IPM (1.01%) | _[ (4.36%) | build (0.97%) | plex (1.04%) | brand (0.78%) | _Quest (0.91%) |
| Block 19 (2.17%) | _and (2.13%) | _cross (3.75%) | _proceeds (5.61%) | _named (1.21%) | _called (0.93%) | _parallel (1.08%) | Shares (0.96%) | _lost (0.81%) |
| Block 20 (2.64%) | , (3.62%) | ": (0.98%) | rons (1.15%) | _Neural (2.26%) | _coupled (4.39%) | _omn (2.30%) | fect (4.73%) | _Fly (1.73%) |
| Block 21 (1.27%) | , (3.47%) | _ft (0.97%) | ysis (1.03%) | _template (1.09%) | _with (0.83%) | _latter (1.09%) | adic (0.79%) | âM¢ (0.87%) |
| Block 22 (3.88%) | , (3.56%) | types (0.98%) | _Turing (2.15%) | . (7.00%) | _which (4.55%) | _most (5.96%) | gress (1.06%) | _VT (5.74%) |
| Block 23 (3.17%) | , (3.95%) | tv (1.07%) | blade (0.96%) | _..." (1.16%) | _j (2.87%) | _model (5.98%) | du (4.83%) | _erg (4.52%) |
| Block 24 (5.36%) | , (3.89%) | _prayers (5.37%) | _Turing (6.05%) | , (6.95%) | _which (5.59%) | _brain (6.37%) | Memory (5.62%) | als (3.00%) |
| Block 25 (2.84%) | , (3.80%) | _complex (0.86%) | _surgery (0.93%) | " (0.97%) | _Neural (1.57%) | _one (5.52%) | _EEG (3.47%) | , (5.60%) |
| Block 26 (5.61%) | , (3.63%) | _dot (6.73%) | _Turing (6.16%) | _for (7.62%) | _then (6.26%) | _Neural (5.36%) | occy (5.16%) | _robot (3.94%) |
| Block 27 (4.91%) | , (3.64%) | ?" (7.12%) | _algorithm (2.21%) | ", (6.61%) | _where (5.86%) | _so (5.87%) | vier (1.80%) | _or (6.21%) |
| Block 28 (3.91%) | , (2.94%) | _solution (0.91%) | _simulation (4.19%) | ", (5.57%) | _which (5.97%) | _F (6.14%) | imil (0.95%) | _Mega (4.63%) |
| Block 29 (4.07%) | , (1.51%) | _life (6.69%) | _network (2.58%) | ) (2.36%) | _using (5.32%) | _neural (6.09%) | Washington (4.30%) | _brains (3.73%) |
| Block 30 (5.05%) | , (1.96%) | Äɬ (5.52%) | _net (5.50%) | _that (7.83%) | _neural (6.24%) | _neural (6.05%) | _underground (4.91%) | _Brain (2.39%) |
| Block 31 (5.02%) | , (2.04%) | "( 6.84%) | _Machine (3.46%) | ," (7.99%) | _neural (6.56%) | _neural (6.10%) | onet (0.95%) | _neural (6.19%) |
| Block 32 (2.06%) | , (2.06%) | ' (5.21%) | _net (0.94%) | ' (7.68%) | _called (6.27%) | _simple (6.34%) | haus (5.11%) | 3 (6.41%) |
| Block 33 (3.65%) | , (2.08%) | ' (0.83%) | _assembly (5.90%) | ' (1.61%) | _to (5.86%) | _TW (1.51%) | Global (5.96%) | _LL (5.41%) |
| Block 34 (2.57%) | , (2.10%) | _to (1.01%) | _vide (0.99%) | , (2.72%) | _and (1.15%) | _class (1.00%) | lc (5.89%) | , (5.73%) |
| Block 35 (1.67%) | , (2.12%) | client (1.09%) | _NET (1.00%) | Ċ (3.33%) | _and (2.74%) | _reservoir (1.16%) | Draft (1.02%) | _scripts (0.93%) |
| Block 36 (1.28%) | ¢ (2.69%) | Ċ (1.06%) | gil (1.03%) | Ċ (1.15%) | Ċ (1.01%) | _Leopard (1.22%) | artist (1.05%) | stals (1.02%) |
| Logits | _ | " | _network | ' | _which | _neural | c | , |
| Expan. (0.980) | , | - | _network | _for | _which | _neural | - | , |

Figure 16: Joint jet lens with learnable weightings ($k = 0$), applied over GPT-2-large with the input sentence "new simple neural architecture, the Transformer"

| | new | _simple | _neural | _architecture | , | _the | _Trans | former |
|---|---|---|---|---|---|---|---|---|
| Block 1 (3.50%) | bie (3.17%) | " (4.75%) | _network (5.93%) | " (3.61%) | _which (1.15%) | _neural (1.60%) | c (5.06%) | _is (2.74%) |
| Block 2 (3.14%) | _ (0.84%) | " (4.15%) | _network (5.49%) | ' (1.80%) | _which (4.28%) | _neural (4.04%) | c (3.60%) | _is (0.93%) |
| Block 3 (1.19%) | _ (0.86%) | " (0.91%) | _network (0.84%) | ' (1.05%) | _which (1.81%) | _neural (2.17%) | c (0.78%) | _is (1.08%) |
| Block 4 (1.08%) | - (0.77%) | ton (1.88%) | _network (1.27%) | ' (0.99%) | _we (0.96%) | _neural (0.94%) | c (0.75%) | _is (1.07%) |
| Block 5 (0.98%) | _ (0.74%) | " (1.03%) | _network (0.98%) | ' (1.06%) | _where (1.01%) | _brain (1.00%) | c (0.88%) | _is (1.13%) |
| Block 6 (1.29%) | _ (3.29%) | " (1.01%) | _network (0.93%) | ' (1.07%) | _and (1.00%) | _neural (1.00%) | c (0.93%) | _is (1.06%) |
| Block 7 (1.32%) | _ (3.60%) | " (1.04%) | _network (0.97%) | ' (1.10%) | _which (1.00%) | _neural (1.00%) | parent (0.89%) | _is (0.97%) |
| Block 8 (1.35%) | _ (3.71%) | " (1.05%) | _network (0.95%) | ' (1.07%) | _which (0.98%) | _researchers (0.99%) | ient (0.97%) | _is (1.10%) |
| Block 9 (1.44%) | , (3.74%) | " (1.04%) | _network (0.83%) | ' (1.07%) | _which (0.99%) | _neural (0.99%) | c (0.94%) | _is (1.91%) |
| Block 10 (1.47%) | - (3.73%) | " (1.04%) | _network (1.44%) | ' (1.07%) | _which (0.97%) | _neural (0.99%) | former (0.93%) | _AI (1.57%) |
| Block 11 (1.36%) | - (3.71%) | " (0.98%) | _network (1.01%) | ' (1.12%) | _which (0.98%) | _neural (0.98%) | c (0.99%) | _is (1.10%) |
| Block 12 (1.36%) | _ (3.69%) | " (1.00%) | _network (1.04%) | ' (1.08%) | _which (0.97%) | _neural (0.97%) | c (1.03%) | , (1.12%) |
| Block 13 (1.35%) | _ (3.65%) | " (1.01%) | _network (1.04%) | " (1.10%) | _where (0.96%) | _neural (0.96%) | c (1.01%) | _Cortex (1.09%) |
| Block 14 (1.31%) | _ (3.61%) | " (1.00%) | _network (1.02%) | ' (1.07%) | _a (0.74%) | _neural (0.92%) | ient (1.00%) | _is (1.10%) |
| Block 15 (1.30%) | _ (3.54%) | " (0.99%) | _network (1.03%) | ' (1.07%) | _which (0.93%) | _neural (0.93%) | c (1.00%) | _chip (0.90%) |
| Block 16 (1.30%) | _ (3.43%) | " (1.04%) | _network (0.95%) | ' (1.09%) | _and (0.89%) | _neural (0.89%) | c (0.99%) | , (1.13%) |
| Block 17 (1.28%) | _ (3.36%) | " (0.97%) | _network (0.95%) | ' (1.09%) | _which (0.90%) | _neural (0.86%) | c (0.99%) | . (1.10%) |
| Block 18 (1.14%) | _ (2.81%) | _ (0.92%) | _network (1.00%) | ' (0.90%) | _a (0.74%) | _more (0.79%) | c (0.90%) | _chip (1.09%) |
| Block 19 (0.99%) | _ (0.98%) | " (0.84%) | _network (0.88%) | ' (0.95%) | _or (1.44%) | _neural (0.76%) | c (0.98%) | _architecture (1.10%) |
| Block 20 (1.53%) | , (0.95%) | x (0.88%) | _network (0.95%) | ' (0.99%) | _we (3.52%) | _authors (3.11%) | c (0.77%) | _is (1.07%) |
| Block 21 (1.23%) | , (0.96%) | " (0.86%) | _networks (0.90%) | ' (1.04%) | _neural (1.93%) | _network (1.16%) | c (1.93%) | _is (1.07%) |
| Block 22 (1.92%) | - (0.96%) | " (2.47%) | _network (0.88%) | ' (1.05%) | _we (4.10%) | _neural (4.13%) | c (0.78%) | _Brain (0.98%) |
| Block 23 (2.10%) | _ (0.90%) | _stuff (0.79%) | _network (1.16%) | ' (0.85%) | _similar (3.67%) | _cu (4.65%) | c (3.79%) | _is (0.99%) |
| Block 24 (3.00%) | _ (0.93%) | " (2.25%) | _network (4.69%) | ' (2.88%) | ' (4.60%) | _ART (4.85%) | c (2.96%) | , (0.85%) |
| Block 25 (3.99%) | "]=> (3.39%) | ton (4.25%) | _net (2.85%) | ' (2.19%) | _with (4.38%) | _loc (4.88%) | c (5.43%) | _S (4.59%) |
| Block 26 (3.96%) | Instance (3.52%) | ' (3.67%) | _network (3.98%) | ' (4.45%) | _Cooper (4.93%) | _first (4.80%) | c (4.25%) | , (2.07%) |
| Block 27 (4.99%) | _ (3.24%) | tons (5.87%) | _network (4.56%) | _of (5.90%) | _but (4.78%) | _neuron (4.83%) | c (4.85%) | _Memory (5.85%) |
| Block 28 (5.13%) | _ (3.08%) | ton (5.20%) | _network (5.48%) | _for (5.93%) | _NI (4.98%) | _first (4.92%) | ient (5.17%) | _uses (6.28%) |
| Block 29 (5.04%) | _ (3.27%) | me (5.80%) | _network (5.64%) | *. (5.22%) | _NAT (4.95%) | _authors (4.94%) | ient (5.52%) | _3000 (5.00%) |
| Block 30 (4.88%) | _ (3.40%) | _kitchen (4.88%) | _network (5.69%) | " (5.41%) | _prototyp (4.94%) | _algorithm (4.88%) | ient (5.55%) | _uses (4.30%) |
| Block 31 (5.31%) | _ (3.61%) | x (6.06%) | _network (3.85%) | ' (6.79%) | _geared (5.16%) | _traditional (5.00%) | c (5.28%) | _XL (6.76%) |
| Block 32 (5.51%) | - (3.70%) | _white (5.66%) | _network (5.56%) | " (6.48%) | ", (5.09%) | _WS (5.03%) | c (5.33%) | _is (7.26%) |
| Block 33 (5.75%) | , (3.73%) | " (6.05%) | _network (6.01%) | " (6.91%) | _which (5.15%) | _neural (5.05%) | c (5.66%) | _Robot (7.46%) |
| Block 34 (5.88%) | , (3.73%) | ton (6.26%) | _network (6.49%) | *, (6.91%) | _which (5.15%) | _neural (5.04%) | ient (5.96%) | _Cortex (7.50%) |
| Block 35 (5.77%) | - (3.74%) | " (6.11%) | _network (6.26%) | _model (6.90%) | _modeled (5.03%) | _neural (4.97%) | ient (6.03%) | _model (7.17%) |
| Block 36 (5.85%) | _ (3.67%) | " (6.29%) | _network (6.51%) | ' (6.77%) | _which (4.95%) | _neural (5.00%) | c (6.10%) | _is (7.52%) |
| | | | | | | | | |
| Logits | _ | " | _network | ' | _which | _neural | c | , |
| Expan. (0.994) | _ | " | _network | ' | _and | _neural | c | _is |

Figure 17: Joint jet lens with learnable weightings ($k = 1$), applied over GPT-2-large with the input sentence "new simple neural architecture, the Transformer"

