# OpenReview forum: "Jet Expansions of Residual Computation"
_ICLR.cc/2025/Conference — Submitted to ICLR 2025_

### Official Review · Reviewer_Xfnq · 2024-10-30

**Soundness:** 3
**Presentation:** 3
**Contribution:** 3
**Rating:** 6
**Confidence:** 2

**Summary:**

This paper utilizes a convex combination of Taylor expansions to rewrite residual networks up to a nonlinear residual term. Crucially, these expansions are such that the contributions from different combinations of network subunits can be studied separately. This results in a data-independent interpretation tool for understanding black-box residual networks. The authors use this developed tool to study the functionality of subunits, pretraining dynamics, and finetuning in the context of language models.

**Strengths:**

The proposed approach is principled, intuitive and unifies certain prior works. As shown by experiments, it can identify the linguistic functionality of various computational subunits in language models.

**Weaknesses:**

Overall, the approximation quality of the jet expansions is not guaranteed, and hence faithfulness to the actual network and its behavior is unclear. This is acknowledged by the authors, and the approximation quality does not necessarily improve with scaling k (as seen from the experiments). Therefore, experimental explorations with jet expansions are only indicative without any confidence.

**Questions:**

In Figure 2, the truthfulness of jet logits decay when $k=2$. Could you comment about scaling with respect to $k$? Do you expect it to improve the quality of the expansions?

Can you explain how to obtain bi-gram and skip-n-gram expansions? Overall, I think these two paragraphs (L259 "Jet bi-grams and skip-n-grams statistics") could be a bit more detailed for comprehension of the reader.

---

> ### Author Response · Authors · 2024-11-30
>
> Thanks for reviewing our paper and for recognizing our method is principled, intuitive and unifies prior works. We would like to address your concerns as follows.
>
> ---
>
> ###  1. Faithfulness to the actual network
> **Question:  the truthfulness of jet logits decay when k=2. Could you comment about scaling with respect to k? Do you expect it to improve the quality of the expansions?**
>
> Thanks for the question. We do not expect that the quality of the expansion necessarily improves with $k$. There are various reasons for this. The foundational one is that in our setting we cannot directly control how far the variate is from the centers, since both of these (set of) quantities are functions of the input (e.g. of the sentence tokens). In particular the variate is the residual stream at a layer we are expanding. Therefore, for specific choices of centers and inputs, the variate can in principle be “far” from the centers (and the centers can be far from each other). In these cases, likely a lower $k$ may in fact achieve a higher fidelity.
>
> This being said:
> - we believe in many cases, already $k=1$ may bring substantial advantages over $k=0$ as it “unlocks” terms that mix different components.  This is straightforward to see e.g.  for the iterative jet lens:
> $$
> J^1 \upsilon(h_l(z))(h_L) = \upsilon(h_l(z)) + U \mathrm{D}\gamma_{L+1} (h_L(z) - h_l(z))
> $$
> Please also see  Figure 3 of the new revision, where we compare with the LogitLense (= iterative jet lens with k=0)
> - We do not seek to “unconditionally approximate” a given network, but rather we apply jets to interpret the network behaviour, seeking to map behaviour to components. In other words, we could not possibly expect that a complex network can be faithfully described by any of its components (in isolation) – and in this sense “it is fine” to obtain an high approximation error if the paths in consideration only capture a small portion of the network (we would be rather surprised if we found otherwise). On the other hand, since a jet path is in effect part of the model, by design it describes a part of the computation of the model. Such part can be more or less relevant, ultimately depending on the input.
>
> ### 2. Expositions on bigram and skip-n-gram expansion
> **Question: Can you explain how to obtain bi-gram and skip-n-gram expansions? Overall, I think these two paragraphs (L259 "Jet bi-grams and skip-n-grams statistics") could be a bit more detailed for comprehension of the reader.**
>
> Thank you for your suggestion. We have revised our paper accordingly based on your feedback. In our new revision, Sec 5.1 describes the experimental setup. We have added a detailed description with  procedures and algorithmic steps for obtaining bi-gram and skip-n-grams in appendix C. Please also consider reading section 3 “Jet expansions and
> -grams” of our reply to reviewer cNS2.
>
> ---
>
> We would like to thank you for your reply again. And please let us know if you have additional questions.

---

### Official Review · Reviewer_cNS2 · 2024-11-01

**Soundness:** 4
**Presentation:** 2
**Contribution:** 3
**Rating:** 5
**Confidence:** 4

**Summary:**

The authors rewrite a model with skip connections as a composition of many Taylor series, evaluated at various points that are related to intermediate activations, promising data-independent global interpretability. The approach is demonstrated on LLMs and conceptually compared to related methods such as LogitLens.

**Strengths:**

- The mathematical exposition is clear.
- The authors acknowledge the limitation of their method in capturing the nonlinear model exactly.
- The applicability of the proposed method for evaluating models globally is interesting and promising. In particular, the model diffing experiments provide the potential of useful metrics for assessing the effectiveness of a specific fine-tuning method, the rate of model improvement and saturation, and the potential for certain emergent properties from n-gram statistics.

**Weaknesses:**

- The first four sections contain clear mathematical expressions. The remaining sections do not use any of these notations which makes it very hard to digest what the figures are measuring in the context of the proposed method. I’d encourage the authors to improve the clarity of the figures and the captions. At the moment, they are very unclear.
- One way to address the first weakness would be to add a new section, between the theoretical section and the empirical section, which explains in the greatest detail possible what exactly is going to be measured empirically.
- The authors claim that their method provides global interpretability but it is unclear how their method is able to provide insights without evaluating the Jacobians at specific points.
- Is the proof of Lemma 1 a novel contribution or is it a well-known result?
- The proof of Lemma 1 (in the Appendix) is not clear.
- Before Equation 9, you state “x_{empty set} = eta”. The notation $x_set$ wasn’t previously defined and is therefore unclear.
- Could the authors expand on the algorithm bubble? At the moment, the steps are not very clear.
- It is unclear from the paper how the jet expansion relates to n-grams.
- The relation between LogitLens and the proposed method should be made more explicit.
- Superposition is mentioned multiple times throughout the paper. The relation with the jet expansion is not clear.

**Questions:**

- "filtering out all paths that involve self-attention modules" — why is this necessary or reasonable?
- Figure 2
    - How is the top table related to the bottom figures?
    - Is it necessary to put all this information in a single figure?
    - Why should we measure the “cosine similarities between original and jet logits of joint (left) and iterative (right) lenses”? What do we learn from measuring it?
    - How do we see evidence for superposition or neuron polysemy in this figure?
    - Is a simpler method like LogitLens capable of identifying similar patterns?  Is there some simpler baseline you could compare your method to?
- Tables 1 and 2
    - "∆ logit after intervention" — what is the exact definition? unclear what this means.
    - What's the order of the expansion?
    - Is a simpler method like LogitLens capable of identifying similar patterns? Is there some simpler baseline you could compare your method to?
- "One-to-one bi-grams like" and “Many-to-many bi-grams” — unclear what does this mean.
- Figure 4
    - What’s the definition of a "hit ratio"?
    - What’s the definition of "total mass"?
    - How do I see double descent or grokking in this figure?
- What’s the definition of “diffing jet bi-grams”?
- "small change in mass" — what’s the definition of “mass”?

---

> ### Author Response · Authors · 2024-11-23
>
> Thank you for reviewing our paper. We appreciate that you found “the proposed method for evaluating models globally interesting and promising” and “the mathematical exposition clear”. We understand that your main concerns relate to the level of details and clarity in the experimental sections. For this, we will upload a revision with better links between the initial sections and the experiments section, with formal definitions of all the quantities we have considered in the experiments. In the meantime, we address here some possible misunderstandings and provide a series of clarifications.
>
> ---
>
> ### 1. Global interpretability
> **Question: The authors claim that their method provides global interpretability but it is unclear how their method is able to provide insights without evaluating the Jacobians at specific points.**
>
> You are correct in noting that we can compute jets (and hence expansions) at specific points and interpret individual examples, as done in our sec 5.1 and in many example-centric interpretability tools, such as LogitLens.
>
> To move beyond example-centric interpretability -- where we focus on understanding individual examples -- towards global interpretability, we observe that the input/output space of a transformer ``body’’ can be fully described with its embeddings and unembeddings, which are inherent parts of the transformer model itself. In other words, one can think of a transformer as an advanced system for computing interactions between embeddings and unembeddings. The strongest of these interactions provide insight into the model's global behavior. For instance, the relationship between 'ing' and 'play' reflects a frequent and significant interaction.
>
> Our approach relies on exhaustively computing and ranking these embedding-unembedding interactions to provide global interpretability, eliminating the need to evaluate individual examples from an external dataset. Jet expansions allow us to do this by  “restricting the model” to simpler interactions, e.g. the bigram interactions and their relevant computational paths. These restricted models are far smaller and more atomic than the original model, making it feasible to compute the expansions exhaustively across the entire (restricted) input-output space, since the number of possible input/output token pairs is finite for the restricted interactions. In this light, jet expansions allow you to understand how embeddings and unembeddings interact within the model, approximating how the model behaves over its entire (restricted) input space rather than approximating how it behaves over a particular point in the input space.
>
> ---
>
> ### 2. Expositions on the method
> **Question: Lemma 1, novelty and proof.**
>
> To the best of our knowledge, this is a novel contribution. Regarding the proof, the intuition is that you can expand a function around any center $x_i$ (separately), treating each time the sum of all points $x_i$ as the variate $y$. Then, you can “recompose” the various parts by taking a convex combination of these various expansions (in fact, even a linear combination would do), and simply observing that $\sum w_i f(\sum x_i) = f(\sum_i x_i)$ for any w that sums to 1. We are happy to improve the clarity of the proof, please let us know which passage is unclear.
>
>
> **Question: Before Equation 9, you state “x_{empty set} = eta”. The notation wasn’t previously defined and is therefore unclear.**
>
> This notation, introduced at the referenced line, is self-contained and does not rely on prior definitions or context. However, if it aids clarity, we could replace the $=$ symbol with $\coloneqq$ to emphasize that this is a definition.
>
> By using  this notation we intend to
>
> A) Establish a connection with the linear case, where the set in the subscript indicates which specific non-linear blocks the path passes through. For example: $x_{\{1\}}$ indicates the path passes through the first nonlinear block; $x_{\{1, 2\}}$ indicates it passes through the first and second nonlinear blocks;
> $x_\emptyset$ indicates no nonlinear blocks are traversed, and the path directly feeds embeddings to the decoder.
>
> B) Provide a succinct description of Eq. (10), which parallels and “mimics” the structure of Eq. (5) from the linear case, as noted in line 196.
>
> **Question: Could the authors expand on the algorithm bubble? At the moment, the steps are not very clear.**
>
> The key idea is that **Algorithm 1** performs local expansion around specified user-provided centers for a single layer, while **Algorithm 2** uses it repeatedly to expand the entire network into exponentially many paths. We explain the two algorithms in detail as follows.

---

> ### Author Response · Authors · 2024-11-23
>
> For **Algorithm 1**:
>
> 1. **Line 1**: Perform the expansion of the $l+1$th non-linearity (i.e. either a block non-linearity, or the final decoder non-linearity) around each center, following Lemma 1\.
> 2. **Line 3**: For a block non-linearity, add the expansion of the identity (due to the residual connection)
> 3. **Line 4 and 5**: Computes the remainder (a function of the input and jet weights) as the difference between the expansion and the actual computation
>
> For **Algorithm 2**:
>
> 1. **Line 1**: Initialize the exponential expansion with the embedding module and the first block non-linearity
> 2. **Line 3**: Recursively call the \`jet\_expand\` algorithm taking as centers the previously obtained expansions. Note that because of the residual connections, the previous expansions are “carried over” through the identity (line 3\. of jet\_expand algorithm), and are joined by the “new expansions” obtained at line 1 of jet\_expand. In this way, at each iteration, the size of $\\xi$ grows by a factor of 2\.
> 3. **Line 4**: Fixes the jet weights to be uniform, for simplicity.
>
> We hope this clarifies the steps of the algorithms. Please let us know if anything is still unclear.
>
> ---
>
> ### 3. Jet expansions and $n$-grams
>
> **Question: It is unclear from the paper how the jet expansion relates to n-grams.**
>
> Thank you for the question. We acknowledge that the explanation in the paper is somewhat condensed and appreciate the opportunity to clarify further. We will incorporate the following discussion into revision.
>
> **General Concept of $n$-Gram Models** $n$-gram models linked to (transformer-based) LMs involves defining or constructing mappings that *functionally* *depend* only on $n-1$ input tokens (with the $n$-th token being the output token) to capture and describe the behavior of the LM. We are not the first to explore this idea; for instance \[3\] fits n-grams on the same dataset used to train the LM; (see lines 490-495).
>
> **Jet Expansions for In-Model $n$-Grams** Jet expansions allow us to define $n$-grams that are derived solely and directly from the model itself -- producing *in-model* $n$-grams rather than *in-data* $n$-grams. This approach offers at least two advantages:
>
> 1. No Dataset Preparation: It does not require preparing datasets for collecting activation patterns when interpreting the model globally, thus saving both time and compute. The process can be performed on CPU, which is about 10 times cheaper per hour than GPUs in the current market.
> 2.  Avoidance of Fitting Artifacts: It circumvents artifacts that may arise from the choice of an external $n$-gram fitting method.
>
> **Jet bi-grams** are paths that **do not pass through self-attention** (self-attention will look at tokens in the context instead of limiting to current token and the output token). In experiments, we focus on two types of bi-gram paths:
>
> 1. the embedding-unembedding path, which can be obtained as jet\_expand(q, L, {η}, 0\)  (see line 289).
> 2. paths that pass through one MLP module. Assuming MLPs occupy odd block indices in the residual network architecture, these expansions can be obtained with the following calls to the jet\_expand algorithm:
>    1. $\mathcal{C} \= \{\eta\} $
>    2. for l=1, 3, … L \-1:
>       1. $\\xi, \\delta$ \= jet\_expand(q, l, $\\{\\eta\\}$, 0\)
>       2. $\\mathcal{C} \= \\mathcal{C} \\cup \\{e(\cdot, 1)\\}\_{e\\in\\xi}$
>    3. $\\xi, \\delta$ \= jet\_expand(q, L, $\\mathcal{C}$, 0\)
>
> This procedure yields a series of functions in $\\xi$ \-- one for each MLP layer \-- that *functionally depend only on the last input token*. Thus, these functions *define* (conditional) bi-grams, once we apply a softmax normalization to their logit output.  Similar constructions can be done for paths that pass through multiple MLPs. We will release code for these, and will also include equivalent algorithms that achieve the same result by calling directly transformer modules.
>
>  **Jet tri-grams** are paths that pass through one self-attention layer. We need also to isolate contribution from the first token of the tri-gram. Concretely, here is the procedure for extracting the 0-th order jet trigram path that passes through the i-th self-attention layer (assuming it has only one head):
>
> 1\. Define $\\sigma\_2(z)=(z\_{t-1}, z\_t)$ as the function that extracts the last two tokens from a sequence of at least length 2\.
>
> 2\. $\\xi, \\delta$ \= jet\_expand$(q, i, \\{\\eta \\circ \\sigma\_2\\}, 0)$
>
> 3\. $\\xi, \\delta$ \= jet\_expand$(q, L, \\{e(\\cdot, 1)\\}\_{e\\in\\xi}, 0)$
>
> This procedure yields a map that functionally depends on two input tokens only, isolating the contribution of the i-th self-attention layer on pairs of tokens, and thus defining a tr-gram (once we apply softmax normalization). Note that this could represent a skip trigram or a contiguous trigram, depending on how the positional information is encoded (e.g. absolute positional embeddings vs rotary embeddings).

---

> ### Author Response · Authors · 2024-11-23
>
> ---
>
> ### 4. Jet expansions and LogitLens
> **Question: The relation between LogitLens and the proposed method should be made more explicit.**
>
> We describe the relation between LogitLens and jet expansion in line 246-258: LogitsLens is, in fact, a special instantiation of jet expansion. We can formalize this as a remark or theorem if you believe it would improve the exposition. Below, we provide an intuitive explanation of their relation.
>
> LogitLens [1,2] projects the representation $h\_l$ at layer $l$ onto unembeddings, effectively performing early decoding of the residual stream. The intuition is that final layer performs the decoding, and since representations are built incrementally across layers, same decoding can be applied iteratively to early layers. We agree with this intuition.
>
> However, we identify a deeper connection between such early decodings and the model’s overall computation. Specifically, early decoding corresponds to isolating the portion of computation up to layer $l$ from the overall computation. Jet expansion formalizes this action of “isolating” and generalizes it to encompass any portion of the model, not just the contribution of $h\_l$. For example, jet expansion allows isolating skipping paths, such as the path going through first layer 1 and 2, skipping layer 3 and going to layer $L$. LogitLens does not support them.
>
> Moreover, since LogitLens is only one instantiation of such "isolating’’ actions, it does not support analysis into the relationship of multiple ``isolating’’ actions. For example, LogitLens can not reveal that a linear transformer can be decomposed into the ensemble of $2^L$ paths, a property explored in [4] when analyzing residual net behavior.
>
> By embedding methods like LogitLens within the analytical framework of jet expansion, we can systematically compare and extend these lens.
>
> Mathematically, LogitLens corresponds to a 0th-order jet expansion applied to $h\_l(z)$ and evaluated with input $z$:
>
> $$\text{LogitLens}_l (z) \= U \gamma(h_l(z)) = J^0 \upsilon (h_l(z)) (h_L(z))$$
>
> This corresponds to calling jet expansion algorithm with the following arguments: jet\_expand$(q, L, \\{h\_l\\}, 0)$.
>
> ---
>
> ### 5. Figures, tables, metrics etc.
> **Question: Superposition is mentioned multiple times. The relation with the jet expansion is not clear.**
>
> We mentioned superposition in Sec 5.1. There, we mean that a single component can take on multiple “roles” and that multiple components may synergize to achieve a specific functional outcome. This makes it challenging to attribute a role, such as “A performs B,” to any single component in isolation. Such entanglement among neural components is also discussed in the literature on superposition and neuron polysemy. Jet expansions with $k>0$ can capture these synergies (e.g. as opposed to logit lens or knock-out ablation). For instance, for $k=1$, we can write iterative jet lens of layer $l$ at $z$ as follows
>
> $$
> J^1 \\upsilon(h\_l(z))(h\_L) \= \\upsilon(h\_l(z)) \+ U \\mathrm{D}\\gamma\_{L+1}(h_l(z)) (h\_L(z) \- h\_l(z))
> $$
>
> The 1st term of the right-hand side is exactly LogitLens, while the 2nd term captures interactions between layers up to $l$ and the layers after $l$.  This interplay indicates that the 1-to-1 mapping between layers and their functions are somewhat limited -- the functions are *superposed* among the layers, and disentangling is needed for interpreting them.
>
> #### Figure 2
> **Question: How is top table related to bottom figures?**
>
> They concern the same type of experiments with jet lenses. In top plots we report one particular case study (sentence); in bottom plots we report quantitative analysis over multiple sentences to show how expansions are faithful to the model output.
>
> **Question: Is it necessary to put all this information in a single figure?**
>
> We included these elements in a single figure due to space constraints. We will separate them into different figures if more space is given in the camera-ready.
>
> **Question: Why should we measure the “cosine similarities between original and jet logits of joint (left) and iterative (right) lenses”**
>
> We measure cosine similarities to assess the faithfulness of the jet lens outputs to the original model computations -- the higher similarity, the more faithful the lens are. Specifically, we measure two things 1) cosine similarities between original and jet lens of the joint variant (left) 2) cosine similarities between original and jet lens of the iterative variant.
>
> **Question: Is a simpler method like LogitLens capable of identifying similar patterns?**
>
> LogitLens is indeed a relevant baseline and is explicitly *included* as a special case of our method in the iterative jet lens framework, where the jet order k=0. In figure 2 bottom right, LogitLens are the solid lines (k=0). We can see that k=1 is more faithful to the model (as measured by cosine similarity). We will revise the paper to make this clear.

---

> ### Author Response · Authors · 2024-11-23
>
> #### Tables 1 and 2
>
> **Question: ∆ logit after intervention**
>
> We apologize for the confusion. To compute \(\Delta\) logits, we calculate the logits for the given \(n\)-gram both before and after applying the intervention, then determine the change in the logits.
>
> For example, in Table 2, consider the trigram *("Lemma, let, s")*. We compute the logit of "s" conditioned on the input *"Lemma let"*. The intervention involves removing the corresponding attention head (e.g., head 2), and we then measure the change in the logit for "s" as a result of this intervention.
>
> **Question: What's the order of the expansion?**
>
> The order is k=0. We mentioned it in line 289\.
>
> **Question: Is a simpler method like LogitLens capable of identifying similar patterns? Is there some simpler baseline you could compare your method to?**
>
> We appreciate the question and are considering which type of “simpler baselines” may provide similar information. First, we would like to clarify that LogitLens, as remarked above, is a special case of a jet expansion. Therefore, our framework is a strict (and vast) extension over it. LogitLens offers a type of local (example-centric) interpretability of the “evolution’’ in the residual stream. This is indeed what we study in Section 5.1-”Jet lenses”. As you can see e.g. confronting Figures 5 and 6 in the Appendix, the LogitLens might not yield any meaningful insight for models such as GPTNeo 2.7B (this was also noted by the original author of LoigtLens). Therefore, for what concerns Sec 5.1-”Jet lenses”, LogitLens is indeed a baseline we used and we believe we have shown the advantage of considering higher-order lenses.
>
> Instead, Table 1 and 2 are obtained analyzing jet paths that pass through single components, exhaustively computed over all their respective input space (being bi/tri-grams). We do not see how we could obtain similar information with the LogitLens.
>
> We greatly appreciate it if you could indicate to us any baselines that seems appropriate. We will run them in the rebuttal.
>
> #### Figure 3 and 4
> **Question: One-to-one bi-grams like and Many-to-many bi-grams — unclear what does this mean.**
>
> We apologize for the confusion. We refer to the ground truth bi-gram distribution.
>
> One-to-one bi-bigrams are (approximately) unimodal bi-grams that concentrate all mass on a single token: i.e. given $z\_1$,  $\\mathcal{P}\_{\\mathcal{D}}(z\_2|z\_1) \\approx 1$ and given $z\_2$, $\\mathcal{P}\_{\\mathcal{D}}(z\_1|z\_2) \\approx 1$  for a specific pair of token and close to 0 for all others. In the example in the paper, $z\_1=$’&’, and $z\_2=$”amp’’.  $\mathcal{P}\_{\mathcal{D}}$ is the probability distribution induced by the pre-training data.
>
> With many-to-to many bi-grams we refer to the opposite scenario where both the conditional probabilities are highly multimodal. In the example $z\_1=$”make” and $z\_2=$”sure” we have that many other tokens can succeed $z\_1=$”make” or precede $z\_2=$”sure”.
>
> Hope this clarifies the issue; we will add this in the appendix.
>
> **Question: What’s the definition of a "hit ratio"?**
> The Hit Ratio (HR@n), often referred to as hit rate, is a metric commonly used in ranking tasks. In our context, we treat each checkpoint of the language model as a "ranker" of bigrams. The Hit Ratio measures how effectively the current model checkpoint retrieves high-quality bigrams from the set of all possible bigrams.
>
> To quantify the model's progress, we define the bigrams at the final step as the "good" bigrams and measure how quickly the model approaches these high-quality bigrams. Specifically, we compute the HR@n to evaluate how often the model’s output bigrams match those in the \`\`true’’ top \\(n\\) given by the final step.
>
> Formally, the Hit Ratio@n  is given by:
>
> \\\[
> HR@n \= \\frac{1}{n} \\sum\_{i=1}^{n} \\mathbb{I}(\\text{the i-th bigram output by the current model} \\in \\text{True\\\_Top\\\_n})
> \\\]
>
> Where:
> \- \\(n\\) is the number of top predictions being considered,
> \- \\(\\mathbb{I}\\) is the indicator function that returns 1 if the \\(i\\)-th bigram output by the model is present in the good top \\(n\\) bigrams, and 0 otherwise,
> \- \\(\\text{True\\\_Top\\\_n}\\) represents the set of "good" bigrams, which in our case we use the $n$ top scoring bigrams from the final model step as the True Top n.

---

> ### Author Response · Authors · 2024-11-23
>
> **What’s the definition of "total mass"?**
>
> We use total mass as a metric to measure the cumulative probabilities of bigrams from the top $1K$ bigrams, weighted by an empirical unigram distribution derived from real data. Formally, it is given by
>
> \\\[ \\text{Total Mass} \= \\sum\_{(z\_1, z\_2) \\in \\text{Top-1k}} \\mathcal{P}\_{e\_t}(z\_2 | z\_1) \\mathcal{P}\_{\\mathcal{D}}(z\_1) \\\]
>
> Where
>
> - $e\_t$ is the embedding-unembedding path at the $t$-th pre-training step
> - \\((z\_1, z\_2)\\) are the bigrams being considered,
> - \\( \\mathcal{P}\_{e\_t}(z\_2 | z\_1) \\) is the probability assigned by the model \\(e\_t\\) (the embedding-unembedding path) for the token \\(z\_2\\) given token \\(z\_1\\),
> - \\( \\mathcal{P}\_{\\mathcal{D}}(z\_1) \\) is the probability of \\(z\_1\\) under the empirical distribution \\(\\mathcal{D}\\), which is the unigram probability given by the Infini-gram API \[5\] on Dolma dataset \[6\] (the dataset used to pretrain the model checkpoints).
>
> This metric is designed to evaluate how much "correct" probability mass the model checkpoints assign to bigrams \\((z\_1, z\_2)\\), taking into account the empirical unigram probability of \\(z\_1\\). It provides insight into how well the model aligns with the empirical distribution of real-world data during the pretraining process.
>
> **Question: How do I see double descent or grokking in this figure?**
> What we meant is that in the plots in Figure 4 we can observe that bi-grams learning dynamics are active throughout the training procedure, even after the training loss stabilizes. This indicates that there is significant behavior change in the model which is not well captured by the training loss, an observation (namely that the training loss is not a good indicator for monitoring model learning progress) that is studied also in grokking and double-descent. In other words, jet bi-grams may offer another point of view for analyzing the learning dynamics compared to pretraining loss.
>
> **Question: What’s the definition of “diffing jet bi-grams”?**
>
> We derive the top-K bi-grams for each model from their embedding-unembedding path, which can be obtained as jet\_expand$(q, L, {η}, 0\)$  (see line 289) \. These bigrams are then saved into CSVs, allowing us to represent models via their respective bigram files. By comparing these files directly, much like comparing text files, we bypass the challenges of comparing the models in the parameter space, where measuring behavioral-level differences can be difficult.
>
> Specially, in your mentioned case, line 449  “diffing jet bi-grams extracted from ...”, we extract the bigram files for Llama-2-7B,  and its coding finetuned versions. By analyzing the resulting bigram files, we can observe differences in behavior by identifying which bigrams are ranked higher in each model. Table 5 in line 824 illustrates this: coding-finetuned versions indeed rank programming-relevant keywords higher. In summary, by transforming models into bigram files (Model → Bigram File), we can effectively compare their behavior via bigram file differences (Model Diff → Bigram File Diff).
>
> We will include a demonstration featuring a UI for comparing jet bigrams in the revision.
>
> ---
>
> Thanks for reviewing our paper again. We hope that the above comments help clarify your doubts and reconsider our submission under a different light.  We understand the rebuttal is somewhat lengthy, so please feel free to reach out if you have any additional questions or if there are specific points we may have missed. We would be happy to provide further explanations or individual replies to ensure all your concerns are addressed.
>
> [1] Nostalgebraist, [*logit lens on non-gpt2 models + extensions*](https://colab.research.google.com/drive/1MjdfK2srcerLrAJDRaJQKO0sUiZ-hQtA), 2021
>
> [2] Nostalgebraist, [*Interpreting GPT: The logit lens*](https://www.lesswrong.com/posts/AcKRB8wDp), 2021
>
> [3] Nguyen T. [*Understanding transformers via n-gram statistics*](https://arxiv.org/pdf/2407.12034), 2024
>
> [4] Andreas Veit, Michael Wilber, Serge Belongie. [*Residual Networks Behave Like Ensembles of Relatively Shallow Networks*](https://arxiv.org/abs/1605.06431), NeurIPS 2016
>
> [5] Soldaini et al. [*Dolma: An Open Corpus of Three Trillion Tokens for Language Model Pretraining Research*](https://arxiv.org/abs/2402.00159), 2024.
>
>
> [6] Liu et al. [*Infini-gram: Scaling Unbounded n-gram Language Models to a Trillion Tokens*](https://infini-gram.io/), 2024.

---

> > ### Author Response · Authors · 2024-11-30
> >
> > Dear reviewer,
> >
> > please note we have updated a revised version of the PDF incorporating many of your suggestions. Please let us know if you have any remaining concern or question.

---

### Official Review · Reviewer_NZ3B · 2024-11-07

**Soundness:** 3
**Presentation:** 2
**Contribution:** 3
**Rating:** 6
**Confidence:** 3

**Summary:**

The authors develop a method that expands a residual network into the sum of an exponential number of jets (roughly, taylor expansions of different components). Each term is a "path" information takes as it traverses the network. They apply this to interperability,

**Strengths:**

The method itself is very interesting; thinking of a network as a sum-of-paths is very natural and the jet formalism seems to capture it in a nice way.

The authors show that this generalizes  prior work such as the logit lens.

Since the top network architectures are residual, this is applicable to the most common model types.

Interpereting parts of the network is an important and relevant topic.

**Weaknesses:**

- The primary issue is confusing exposition and incomplete details. These make the contributions difficult to asses. I enumerated specifics in the "questions" section.
- The exponential expansion factor means that to analyze a model like LLaMa 405b, one would have 2^118 terms which seems a bit unwieldy
- Presumably, you need ways of computing k-th order jets for network components (the authors don't seem to discuss this), which makes implementation difficult.

**Questions:**

- Paragraph after eq (9): what is gamma_3? it is not defined or used in the figure. This seems important. Is it a hypothetical 3rd block?
- Lemma 1: Unclear where w comes from. I have 3 hypotheses: (1) w is arbitrary and changes which member of the equivalence class you have (2) there is a specific value of w needed ("there exists a w...") (3) there's a typo and the LHS should be a weighted sum of x. This needs to be explicit.
- What is the point of tthe if l < L in Algorithm 1? Isn't this always true? In the else, how can there be a gamma_{L+1} if there are L layers? where does w come from?

line 264: "For example, bi-grams statistics related to Pq (z2|z1, . . . ) can be computed
by evaluating bi-gram paths, which we can obtain by expanding the LLM with Algorithm 2 and
filtering out all paths that involve self-attention modules."

If you filter out all self-attention paths, aren't the bigrams z1, z2 independent? this needs to be true because self-attention is the only mechanism for a transformer to route information amongst tokens.

How are you optimizing the jet weights? This seems to be very important as you show the weights in Figure 2, but you only briefly mention that it is "done cheaply" without any details. Do you need specific datasets? do you use SGD? How do you actually compute the residual in a way that can be optimized?

How do you represent (computationally) and evaluate the jets? The paper presents them abstractly, which is fine, but in order to compute anything you need to be able to evaluate the k-th order jet for an MLP or self-attention layer.


Minor comments:

In eq (7), I found the notation `J^k f(x) = f(x) + ...` to be a bit confusing; suggest `(J^k f) x = y \mapsto ...`

---

> ### Author Response · Authors · 2024-11-15
>
> We thank you for your time and appreciate that you found our work “relevant”, our method “very interesting”, and “applicable to the most common model types”. We wish to address some possible misunderstandings and provide clarifications to the issues that you raised.
>
> ---
>
> ### 1. Compute k-th order jets
> **Question: How to compute k-th order jets for network components. / How do you represent (computationally) and evaluate the jets? The paper presents them abstractly, which is fine, but in order to compute anything you need to be able to evaluate the k-th order jet for an MLP or self-attention layer.**
>
> We want to emphasize that the derivatives (jets) in our work are taken with respect to the intermediate input, not the model parameters as it is typically done during training. Specifically, we compute the jets with respect to the vectors that accumulate in the residual stream during inference, when the model parameters are fixed. This choice has significant implications for computational complexity, especially when the number of model parameters vastly exceeds the dimensionality of the residual vector. For example, in Llama-2-7B, the total number of parameters is $7B$ while the residual vector size is just $4096$; a second-order jet w.r.t parameters would take $O(7B^2)$ which is approximately $O(4.9 \times 10^{19})$ memory complexity, while second-order jet  w.r.t a residual vector would take just $O(4096^2)$, around  $O(1.7 \times10^7) $ memory complexity; This illustrates a roughly $12$ orders-of-magnitude difference in memory complexity between the two cases.
>
> By choosing to expand jets on residual vectors rather than model parameters, *we align the expansion process with the structure of recursive residual networks*. This approach naturally tracks the effects of residual vectors as they walk through different paths in the network, making it a computationally efficient and conceptually fitting method for such architectures. Our method is not designed for general neural architectures without recursive residual links, which would be hard to decompose into paths and compute the jets over the paths.
>
> Implementation-wise, the main difficulty in evaluating jets is to computing (higher order) jacobian-vector products (instead of the jacobian itself) and making sure that we differentiate w.r.t the **correct** variables (i.e. the block inputs, rather than block parameters as normally done when training a neural net). This can be done with modern automatic differentiation tools such as `pytorch.autograd` in a straightforward way using the jvp API \[1\]. Here is our pytorch implementation:
> ```python
> def jet(f: Callable[[torch.Tensor], torch.Tensor], x: torch.Tensor, y: torch.Tensor, k: int):
>    """
>    Computes J^f(x)(y); see Eq. (7). This is equivalent to the truncated Taylor expansion of f
>    around x of order k, evaluated at y.
>
>    :param f: a callable (to be evaluated on x).
>    :param x: center, in the domain of f
>    :param y: variate in the domain of f
>    :param k: jet order >= 0
>
>    :return: jet^k f(x)(y).
>    """
>    assert callable(f), "f needs to be a callable function"
>    assert k >= 0, "the jet order needs to be non-negative"
>    res, funcs = [f(x)], [f] + [None]*k
>    yp = (y - x).detach()  # prevents gradient propagation through the variate
>
>    def make_functional(i): # using wrapper to return a function
>        def _jvp_y(_x):
>            # note: jvp returns the tuple  (f(x), D f(x) v), we need the second entry
>            return torch.autograd.functional.jvp(funcs[i], _x, yp, create_graph=True, strict=False)[1]
>        return _jvp_y
>
>    for j in range(1, k + 1):
>        funcs[j] = make_functional(j - 1)  # j-th order is done by taking derivative of the (j-1)-th order
>        d_prev = funcs[j](x)  # eval at x
>        res.append(d_prev / torch.math.factorial(j))  # taylor coefficients 1/j!
>    return sum(res)  # this is the numerical value of the full jet
> ```
> We use this primitive function to implement the expansion algorithm. An example of computing higher-order jets with the above `jet` function can be as follows
> ```python
> # Example function: f(x) = x^2
> def f(x):
>    return x ** 2
>
> x = torch.tensor([2.0], requires_grad=True) # center point
> y = torch.tensor([3.0]) # variate
> k = 2 # order of expansion
> # Compute the jet of order 2 at y
> result = jet(f, x, y, k)
> ```
> We will open-source our code upon acceptance.

---

> > ### Author Response · Authors · 2024-11-15
> >
> > ### 2. Number of paths for analyzing
> > **Question: The exponential expansion factor means that to analyze a model like LLaMa 405b, one would have 2^118 terms which seems a bit unwieldy.**
> >
> > We agree that it is not practical to compute (or examine) all the exponential number of paths in a large model at the size of Llama 405B. Rather, the exponential expansion represents the finest level of granularity we discuss in our work. We think it is worthwhile considering this level theoretically, especially in light of previous literature on analyzing residual network’s behavior (see \[2\] Veit et al. 2016). This maximal expansion in theory demonstrates the flexibility of jet expansions, and allows us to ground our contribution in a more formalized setting.
> >
> > In practice, however, one can choose any \`\`interesting’’ path they want to analyze or merge multiple paths instead of expanding at the finest level. In fact, we speculate that shorter paths are more relevant for understanding global model behavior, possibly extending with the findings reported by \[2\] Veit et al. 2016 in their section 5\. Additional evidence supporting the importance of some paths over others comes from latest research on early exiting \[3\], layer skipping \[4\], where skipping later computation seems not to negatively impact or in some cases even improves the performance of the recursive residual computation. Jet expansion allows users to focus on the most relevant paths for a given target sentence, “skipping” the uninteresting ones.
> >
> > ### 3 Expositions on the decoding nonlinearity $\gamma_{L+1}$, jet weights $w$ in the algorithm
> > **Question: Paragraph after eq (9): what is gamma\_3?**
> >
> > This is the decoding module non-linearity. In the previous equations of the paragraph “Exponential expansion of a two-blocks network” (line 146), $\gamma\_3$ was wrapped in the decoding module $\upsilon$. This is consistent with the notation we introduced in Section 2, please refer to Eq. (3) in line 085. In any case we will make this explicit also in the paragraph in our revision.
> >
> > **Question: Lemma 1: Unclear where w comes from. I have 3 hypotheses: (1) w is arbitrary and changes which member of the equivalence class you have (2) there is a specific value of w needed ("there exists a w...") (3) there's a typo and the LHS should be a weighted sum of x. This needs to be explicit.**
> >
> > (1) is correct. The jet weight $w$ is an arbitrary vector in the $N-1$ dimensional simplex, as it is declared in the Lemma. This means that the equivalence holds for every $w \in \Delta^{N-1}$. The intuition here is that, for every center $x\_i$, you can expand the function by treating the non-chosen points $x\_j$ (where $j\\neq i$) as part of the variate. Then, you can “recompose” the model computation using a convex combination of these different expansions (in fact, even a linear combination would suffice, please refer to Appendix A). The key observation here is that the original function $f(z)$ can be expressed as a weighted sum of multiple versions of itself: $f(z) \= \\sum w\_i f(z)$, where $w\_i$ are the weights associated with the i-th expansion. This is the passage where the jet weights appear. Please see Appendix A for a full proof.
> >
> > **Question: What is the point of the if l \< L in Algorithm 1? Isn't this always true? In the else, how can there be a gamma\_{L+1} if there are L layers?**
> >
> > No, you can call the algorithm with $l=L$, which corresponds to expanding the decoding layer; hence the branch-else in the algorithm (also, please note that the decoding layer is not in the recursive residual stacking). According to our notation in section 2 line 085, $\gamma\_{L+1}$ is the final normalization wrapped in the decoding module $\\upsilon$.
> >
> > **Question: where does w come from \[in the algorithm\]?**
> >
> > The set of expansions $\\xi$’s and the remainder $\\delta$ are both functions that take as input a jet weight vector. So the $w$ must be provided by the user when *evaluating the expansions* (indeed, please note that the algorithm returns functions, not points). Please refer to lines 201-206 for the formal definitions.

---

> ### Author Response · Authors · 2024-11-15
>
> ### 4. Dependence among tokens
> **Question: If you filter out all self-attention paths, aren't the bigrams z1, z2 independent? this needs to be true because self-attention is the only mechanism for a transformer to route information amongst tokens.**
>
>  No, filtering out all the self-attentions still maintains the dependence on the last input token. To clarify, the $z\_2$ represents the output token, while the $z\_1$ is the last token of the input sequence. Perhaps an easy way to convince you about this is to think about a “transformer” with 0 blocks, which is just a special case of the setting we consider. In this case you only have an encoder and a decoder layer, no self-attentions involved. And for tokens (i.e. integers) $z\_1$, $z\_2$ you have
>
> $$\\mathcal{P}\_q(z\_2 | z\_1) \= \\mathrm{Softmax}\[q(z\_1)\]\_{z\_2}$$
>
> where $q(z\_1)=U \\gamma\_1(\\eta(z\_1))$ and the softmax score is indexed with $z\_2$.
>
> So, the paths that do not “pass through” any self-attention still maintain dependence on the last token of the sequence. Hence, they contribute toward the bi-gram statistics of the model.
>
> ### 5. Jet weight optimization
> **Question: How are you optimizing the jet weights?**
>
> We optimized jet weights by minimizing the loss we introduced in Remark 3\. In our experiments we used SGD indeed, but we believe that there is also a closed form solution obtainable via KKT conditions \[5\]  although we haven’t computed nor implemented this yet. Note that the loss indeed depends on an input $z$ as it requires computing the remainder, and hence the expansions. In the experiment of Figure 2 top the input is the sentence of which we are computing the jet lens. We do not use any other datasets except the query sentence itself. We will add details in the appendix (and the optimization routine will be part of the release code).
>
> ---
>
> Thanks for reviewing our paper again. We hope that the above comments help clarify your doubts and reconsider our submission under a different light. Please let us know if you have any further questions\!
>
> [1] **PyTorch Documentation**. [*`torch.autograd.functional.jvp` — Jacobian-Vector Products (JVP) Function*](https://pytorch.org/docs/stable/generated/torch.autograd.functional.jvp.html)
>
> [2] **Veit et al** (2016). [*Residual Networks Behave Like Ensembles of Relatively Shallow Networks*.  NeurIPS 2016](https://arxiv.org/pdf/1605.06431)
>
> [3] **Elhoushi et al** (2024). [*LayerSkip: Enabling Early Exit Inference and Self-Speculative Decoding*. ACL 2024](https://aclanthology.org/2024.acl-long.681)
>
> [4] **Fan et al** (2024). [*Not All Layers of LLMs Are Necessary During Inference*](https://arxiv.org/abs/2403.02181)
>
> [5] [*Karush–Kuhn–Tucker Conditions* ](https://en.wikipedia.org/wiki/Karush%E2%80%93Kuhn%E2%80%93Tucker_conditions)

---

> ### Comment · Reviewer_NZ3B · 2024-11-26
>
> > 1. Compute k-th order jets
>
> yes, i understand that you aren't differentiating w.r.t. parameters here. This calculation method is reasonable, but it’s not obvious from the paper that this is how you are doing it. I think the paper should call out explicitly (in the main body) that this calculation is done by iterating JVP with autograd (code/psuedocode is probably unnecessary, maybe a recurrence equation would be helpful), since this is a very important detail. If that is mentioned explicitly, then I apologize for missing it. It is also important to discuss the complexity of the computation as a function of the order. While I don't have any problem with the abstract presentation, it needs to be connected to the concrete implementation for the method and experiments to make sense together.
>
> > 2. Number of paths for analyzing
>
> This makes sense, but is not obvious (to me) or discussed in the paper (to my recollection). It’s also unclear how one chooses “interesting” paths in the first place; I would think that the jet expansion is the method to find them, no?
>
> > 3. Expositions on the decoding nonlinearity
>
> Thanks for the clarifications.
>
> Regarding the definition of w and its relation to equivalence classes, this is very not obvious from the writing and one should not need to delve into the appendix to understand it.
>
> How w was obtained in the experiments needs to be discussed as well, since presumably they may change the numerical output in some way when the series is truncated (this is implied by remark 3 and your response “5.”).
>
> > 4.
>
>  I see; this implies that this only works for consecutive tokens and does not generalize beyond bigrams, is that correct? I guess you could generalize by filtering out specific subsets of self-attention paths, but again this isn’t really obvious from the text.
>
> > 5.
>
> Thanks for the clarification. The text should make this more clear, perhaps mentioning in the experiment section.

---

> ### Comment · Reviewer_NZ3B · 2024-11-27
>
> I've increased my score based on feedback

---

> ### Author Response · Authors · 2024-11-30
>
> Thank you for responding to our rebuttal and for your valuable suggestions. We have incorporated many of your suggestions in the revised PDF. We address your follow-up questions them as follows.
>
> ---
>
> ### 1. Compute k-th order jets
> **Question: I think the paper should call out explicitly (in the main body) that this calculation is done by iterating JVP with autograd (code/psuedocode is probably unnecessary, maybe a recurrence equation would be helpful)**
>
> Thanks for your suggestion. We have added a recurrence equation and explained the jvp usage in line 202 in our new pdf.
>
> **Question: It is also important to discuss the complexity of the computation as a function of the order.**
>
> Thank you for your suggestion. We discuss the complexity as a function of the order in line 200. Additionally, we have run a new experiment where we report the runtime scaling with the jet order k. We mentioned it in line 205 and the plot is in appendix B in our new pdf.
>
> ### 2. Number of paths for analyzing
> **Question: This makes sense, but is not obvious (to me) or discussed in the paper (to my recollection). It’s also unclear how one chooses “interesting” paths in the first place; I would think that the jet expansion is the method to find them, no?**
>
> Thank you for the feedback. We will integrate the discussion into the final version.
> We do agree with you that an important area of research is about developing automatic methods to find the “interesting” paths. In fact, there has been substantial good work in this space, for example ACDC [1] and "information route" [2], where important circuits or rather paths, are automatically found for given input sentences. Our framework can be used to examine how faithful the extracted path is to the original model prediction -- complementing these works rather than replacing them, and allows to potentially obtain more faithful paths through higher-order jets.  We can find the interesting paths by ranking the faithfulness of each path for given inputs. We will definitely try this out in the future along with many other applications opened by our framework.
>
> However, we would like to stress that this is not the primary focus of this work. This work is mainly for presenting a unified and general framework for taking partial computation out of the entire residual computation graph, which is a common operation when interpreting models. As you recognise in your review jet expansions offer a natural and broadly-applicable framework to achieve this and substantially increases the capacity to extract paths from what was previously achieved by Veit et al., 2016  [3] (i.e. $L+1$ paths, for a network of depth $L$).
>
>
> ### 3. Expositions on the decoding nonlinearity
> **Question: Regarding the definition of w and its relation to equivalence classes, this is very not obvious from the writing and one should not need to delve into the appendix to understand it. How w was obtained in the experiments needs to be discussed as well**
>
> Thank you for your suggestion. We have revised the statement of Lemma 1 accordingly. In We also further specified how we optimize $w$ in Remark 2 and in the experimental setup of  Sec 5.1.
>
> ### 4. Dependence among tokens
> **Question: I see; this implies that this only works for consecutive tokens and does not generalize beyond bigrams, is that correct?**
>
> Yes, you are correct. For the paths that filter out self-attention (e.g. the path that only goes through MLPs), we can only extract jet bi-grams.
>
> **Question: I guess you could generalize by filtering out specific subsets of self-attention paths, but again this isn’t really obvious from the text.**
>
> Yes, you are very correct here that we can filter specific subsets of self-attention paths. Indeed, for our experiments on jet trigrams, we filter out the set to incorporate one specific attention head so that we can see their role, as shown in our Table 2, and there we can see non-consecutive tokens. For example, the tri-gram (_Lemma, _let, _s), the token _Lemma does not necessary directly precede the token _let. Indeed the standard way of writing Lemmas has the indexing and the name of the lemma directly following _Lemma, as in ``Lemma 1 (convex combinations of jets). … Let s …’’
> We describe how we can extract tri-trigrams (which generalizes to larger n) in Appendix C.
>
> ### 5. Jet weight optimization
>
> **Question: Thanks for the clarification. The text should make this more clear, perhaps mentioning in the experiment section.**
>
> Thank you for your suggestion. We have incorporated the suggested changes in Sec 5.1 setup in our new pdf
>
> ---
>
> Thank you for reviewing our paper again. We appreciate your efforts in going to the details of our paper and offer very useful suggestions. We believe, after incorporating your feedback, the revised version (the new pdf) is now more accessible. We hope this version will help you reconsider our work in a new light. In the meanwhile, please let us know if you have more concerns.

---

> > ### Author Response · Authors · 2024-11-30
> >
> > [1] **ACDC, Conmy et al** (2023), [*Towards Automated Circuit Discovery for Mechanistic Interpretability*. NeurIPS 2023](https://arxiv.org/abs/2304.14997)
> >
> > [2] **Information route, Ferrando et al** (2024), [*Information Flow Routes: Automatically Interpreting Language Models at Scale*. EMNLP 2024](https://arxiv.org/abs/2403.00824)
> >
> > [3] **Veit et al** (2016). [*Residual Networks Behave Like Ensembles of Relatively Shallow Networks*.  NeurIPS 2016](https://arxiv.org/pdf/1605.06431)

---

> > > ### Comment · Reviewer_NZ3B · 2024-12-02
> > >
> > > Thanks for the response. I will take a look at the revised pdf.

---

> > > > ### Author Response · Authors · 2024-12-03
> > > >
> > > > Thank you for your feedback and for taking the time to review the revised PDF.

---

### Official Review · Reviewer_S4JS · 2024-11-11

**Soundness:** 3
**Presentation:** 4
**Contribution:** 3
**Rating:** 6
**Confidence:** 4

**Summary:**

The paper introduces a method for expanding residual networks, such as Transformers, using jets—operators that generalize truncated Taylor series. This approach aims to disentangle the contributions of individual computational paths to model predictions. The authors claim that their method subsumes the Logit Lens and demonstrate its ability to extract n-gram statistics from intermediate model layers, also enabling a data-free approach to detoxification.

**Strengths:**

Well-written and theoretically well-developed, with content that is thorough yet not overwhelming.

**Weaknesses:**

I find the theoretical foundation to be solid, but my main concerns lie with the experimental approach.

The experiments may be overly empirical and lack statistical rigor. For instance, in Section 5.1, only a handful of jet paths corresponding to specific linguistic functions are selected to demonstrate intervention effects. A more systematic approach is needed to demonstrate that the jet lens is more effective than the logit lens.

While the paper primarily offers an analytical framework, it lacks actionable insights for model steering. For example, could it be demonstrated that the bi-gram statistics can be leveraged to guide more efficient pre-training or improve RLHF techniques for reducing toxicity?

**Questions:**

I have mostly raised them in the weakness section.

---

> ### Author Response · Authors · 2024-11-30
>
> Thank you for reviewing our paper and for recognizing that our paper is theoretically solid. We would like to address your concerns as follows.
>
> ---
>
> ### 1. Scope
> **Question: While the paper primarily offers an analytical framework, it lacks actionable insights for model steering. For example, could it be demonstrated that the bi-gram statistics can be leveraged to guide more efficient pre-training or improve RLHF techniques for reducing toxicity?**
>
> Thank you for the question. These are indeed very interesting and promising future directions for research, which we will focus on in future work. In this work, our main focus (and contribution) is to introduce the jet expansion framework, as a grounded and unifying analytical framework for expanding a residual net into its constituting paths (up to a super-exponential number of paths, improving on previous work by Veit et al. 2016).
>
> Now with our tools, we can examine the growing number of existing LLMs in the market in a more scalable way (without external datasets and GPUs) and describe them using their corresponding global bi-grams/tri-grams. Such post-training diagnosis toolkit complements the efforts that focus instead on the training method. As LLMs become more powerful and impact the public, our tool can be useful for people that do not train the LLM themselves (and very likely do not have their own GPUs) to peek into what is inside the LLMs and for institutional actors to audit LLMs in a systemic way.
>
>
> ### 2. Jet Paths
> **Question: For instance, in Section 5.1, only a handful of jet paths corresponding to specific linguistic functions are selected to demonstrate intervention effects.**
>
> Thanks for pointing out this. We apologize for the confusion. We report the jet paths passing through MLPs or self-attention heads that in our analysis surfaced clear compelling behaviour. This means that for non-reported paths, we did not readily observe evident behaviour. As we clarified in the appendix of the revised PDF,  our analysis consisted in sorting (conditional) bi/tri-grams by their logit score and picking and analyzing the top scores. As an example, here below we report top-10 bi-grams from Llama2-7B  that pass through MLP 3 (no discernible pattern, not reported) and 6 (adding ‘ing’ pattern, reported).
> #### MLP 3:
> | Rank | Token 1   | Token 2   |
> |------|-----------|-----------|
> | 0    |           |           |
> | 1    | ▁like     | wise      |
> | 2    | ▁&        | amp       |
> | 3    | ▁of       | ▁course   |
> | 4    | ▁&        | quot      |
> | 5    | ▁with     | draw      |
> | 6    | ▁Like     | wise      |
> | 7    | ▁presso   | rez       |
> | 8    | ▁fus      | sche      |
> | 9    | ▁With     | draw      |
>
> #### MLP 6:
> | Rank | Token 1    | Token 2       |
> |------|------------|---------------|
> | 0    | ▁occurr    | ing           |
> | 1    | ▁maintain  | ing           |
> | 2    | ▁arriv     | ing           |
> | 3    | develop    | ing           |
> | 4    | ▁analyz    | ing           |
> | 5    | Work       | ing           |
> | 6    | ▁occurr    | ed            |
> | 7    | für        | ▁Хронологија  |
> | 8    | ▁Fight     | ing           |
> | 9    | ▁or        | anges         |
>
> Please note that:
>  -  We do not claim that any MLP/attention head has “single roles”, or that a role is “fully” captured by any MLP (path); rather we see evidence that behaviour is distributed across components and single components, which (may) have synergic behavior.
> - We have only analyzed a tiny portion of all paths obtainable through jet expansions, in particular, we limited our study to paths that pass through only one MLP/attention head at a time.
> - We also did not implemented higher-order bi-tri/grams extraction, as this requires a further refinement step in the method (which we mentioned in Section 4, first paragraph) to factorize contributions from higher order terms
>
> We plan to continue our research in these directions.

---

> ### Author Response · Authors · 2024-11-30
>
> ### 3. Jet Lens vs Logit Lens
> **Questions: A more systematic approach is needed to demonstrate that the jet lens is more effective than the logit lens.**
>
> We apologize for the confusion. Jet lenses (a particular type of jet expansion) subsume the LogitLens. We have revised the paper to highlight this in Section 5.2 in the new pdf. In Figure 3 (right plot), the solid lines are LogitLens and the dashed lines are JetLens. We observe that the jet lens is more faithful to the original model.
>
> Finally, please also not that the both bi/tri-gram expansions and iterative/joint jet lenses are particular *different* expansions schemes with different interpretability scopes and aims. The paragraphs of section 5 concern with various schemes. We hope we further clarified this in the newly added Sec 5.1.
>
> ---
>
> We hope this clarifies your concerns. Please let us know if you have any remaining question.

---

> > ### Comment · Reviewer_S4JS · 2024-12-03
> >
> > I appreciate the authors' responses and efforts on improving the manuscript. They mostly addressed my concerns. I would like to  increase my confidence score.

---

> > > ### Author Response · Authors · 2024-12-03
> > >
> > > Thank you for reviewing the revisions. We are happy that the updates have addressed most of your concerns. Your increased confidence in the work is greatly appreciated.

---

### Author Response · Authors · 2024-11-30
**Revised PDF**

Dear reviewers,

We would like to thank you all for your time and effort reviewing our paper. We believe your advice has been very helpful in helping us improve the readability, organization, and presentation of this work. Major changes in the main paper are highlighted in red. In the new revision, we have made the following changes:

1. We added a new experiment, where we measure the runtime scaling w.r.t to the jet order $k$.
2. We added a description on using JVP to compute the higher-order jets
3. We introduced the jet weights more clearly in Lemma 1 and described how we optimized them in more detail in Remark 2 and Sec 5.1 “Setup”
4. In  Remark 3, we clarify that the approximation quality depends on both the jet order and the choice of jet centers. This should make it clear that scaling k only does not necessarily result in lower approximation error.
5. We added a section (Section 5.1, “Setup”) that bridges the theory and the experiments, explaining how we compute jet bi- and tri-grams and introducing metrics that we compute in the experiments. In addition, we added more details and formal definition on these in Appendix C and D, respectively.
6. We added details on jet bigram diffing in appendix E.
7. We revise and clarify experimental comparison with LogitLens, decoupled ex Figure 2 and clarified exposition around line 350.

For your convenience, we uploaded a pdf diff in the supplementary material (although some parts that are highlighted as differences have not actually changed).

We thank all reviewers for the suggestions and modified the manuscript to address the concerns.
Our main contribution is a general theoretical framework to identify the contributions of different residual paths, which most reviewers also pointed out. Meanwhile, we would like to know if the reviewers have any further comments.

---

### Meta-Review · Area_Chair_rTCk · 2024-12-21

**Metareview:**

In this work, the authors introduce a framework for expanding residual networks into linear residual networks using Taylor-type expansions (jets). These jets encode path information through the computational graph which can then be interpreted to better understand higher-order behavior during inference. All reviewers expressed mixed opinions of the work, acknowledging the report is generally well-written (at the sentence level), and clearly shows a substantial amount of thought and development. The general idea is attractive, but the report falters with the details. Much of the "weaknesses" raised seem to stem from a general confusion around many of the developed concepts, including how the jets are computed, how they provide insights, what some of the terminology means, what the baselines are actually capable of and how they are subsumed, how the jets relate to n-grams, and what the jet weights are. There are many other missed details here, but given the novelty of the approach and its potential for a new form of mechanistic interpretability, a clear exposition should receive glowing reviews.

I would recommend that the authors reconsider how the document is structured. Perhaps rather than focusing on the nature of the theoretical framework and broad discussion of its implications, I suggest considering the report from the angle of a practitioner and place practical motivations front and center: first, how does one interpret a network using current approaches (e.g. LogitLens); then, what is a general model for interpretability and how might it be used (linear residual networks), what is the ideal toolkit etc.; and finally, introduce the jets as a natural generalization. Provide examples of computational complexity. All implementation details, including the code segment provided to Reviewer NZ3B, are required for reproducibility, but these can go into further Appendices as needed. However, they should not be in remarks, which can be easily ignored when read.

This is just an example, but regardless, I believe the manuscript is not yet ready, and will require an additional review process once it is reformatted to better suit practitioners.

**Additional Comments On Reviewer Discussion:**

Reviewer S4JS provides a mixed review, appreciating the theoretical foundations, but expressing concerns with the experimental approach, as well as some confusion about the relationship between the logit lens and the jet lens. The authors clarify that the logit lens is a special case of the jet lens, and other properties of the experiments. The reviewer is satisfied with these clarifications.

Reviewer NZ3B expresses confusion over the exposition, concern over the exponential expansion factor and inquires about how jets are computed. The authors provide an excellent response, detailing exactly how jets are computed, why the exponential factor is not too bad, how weights are optimized, and a few other details. The reviewer increases their score based on this feedback, commenting that many of these details are not present in the manuscript. Finally, the reviewer outlines some remaining strengths and weaknesses in the updated version, criticizing some missing aspects of the experiments, missing discussion of path pruning which would make the idea practical, and other missing details.

Reviewer cNS2 criticizes the clarity of the non-mathematical exposition, the lack of detail, expresses confusion over how the method provides global interpretability, inquires about the relationships between the jets and n-grams and LogitLens, and many other questions. The authors answer each of these in turn, providing a lot of detail in their responses. The reviewer does not engage further in the discussion.

Reviewer Xfnq asks about whether the quality of the expansion is expected to improve with the order, and about how bi-gram and skip-n-gram expansions are obtained. The authors respond with further details, but the reviewer does not engage further.

---

### Decision · Program_Chairs · 2025-01-22

Reject